# Identification of the regulatory elements and protein substrates of lysine acetoacetylation

Qianyun Fu[1], Terry Nguyen[1], Bhoj Kumar[2], Parastoo Azadi[2], Y George Zheng[1]*

[1]Department of Pharmaceutical and Biomedical Sciences, College of Pharmacy, University of Georgia, Athens, United States; [2]Complex Carbohydrate Research Center, University of Georgia, Athens, United States

## eLife Assessment

This **useful** study reports a method to detect and analyze a novel post-translational modification, lysine acetoacetylation (Kacac), finding it regulates protein metabolism pathways. The study unveils epigenetic modifiers involved in placing this mark, including key histone acetyltransferases such as p300, and concomitant HDACs, which remove the mark. Proteomic and bioinformatics analysis identified many human proteins with Kacac sites, potentially suggesting broad effects on cellular processes and disease mechanisms. The data presented are **solid** and the study will be of interest to those studying protein and metabolic regulation.

*For correspondence:
yzheng@uga.edu

Competing interest: The authors declare that no competing interests exist.

**Abstract** Short-chain fatty acylations establish connections between cell metabolism and regulatory pathways. Lysine acetoacetylation (Kacac) was recently identified as a new histone mark. However, regulatory elements, substrate proteins, and epigenetic functions of Kacac are not yet fully understood, hindering further in-depth understanding of acetoacetate-modulated (patho)physiological processes. Here, we created a chemo-immunological approach for reliable detection of Kacac, and demonstrated that acetoacetate serves as the primary precursor for histone Kacac. We report the enzymatic addition of the Kacac mark by the acyltransferases GCN5, p300, and PCAF, and its removal by the deacetylase HDAC3. Furthermore, we establish acetoacetyl-CoA synthetase as a key regulator of cellular Kacac levels. A comprehensive proteomic analysis has identified 139 Kacac sites on 85 human proteins. Bioinformatics analysis of Kacac substrates and RNA sequencing data reveal the broad impacts of Kacac on multifaceted cellular processes. These findings unveil pivotal regulatory mechanisms for the acetoacetate-mediated Kacac pathway, opening a new avenue for further investigation into ketone body functions in various pathophysiological states.

## Introduction

Cellular metabolites play crucial roles in the production of bioenergy and the synthesis of biomolecules (*Palm and Thompson, 2017*). In addition to this classical functionality, there is a growing acknowledgment that certain metabolic molecules also exhibit regulatory functions by acting as precursors for post-translational modifications (PTMs) of proteins (*DeBerardinis and Thompson, 2012*). Short-chain fatty acids are abundant metabolites widely present in both prokaryotic and eukaryotic cells (*Koh et al., 2016*). Fatty acylation of lysine residues in proteins, acetylation being the most prominent, has been known as a versatile form of reversible PTMs for their capacity to impart diverse regulatory functions on key cellular processes (*Tan et al., 2011*). Lysine acylation reactions are dependent on their respective short-chain CoAs as cofactors (more precisely termed as cosubstrates) and

are regulated by a finely balanced enzymatic counteraction involving lysine acetyltransferases (KATs) and lysine deacetylases (KDACs) (*Komaniecki and Lin, 2021*; *Sabari et al., 2017*). Beyond lysine acetylation, in recent years, multiple fatty acylations on nuclear histones, for example, lysine β-hydroxybutyrylation (Kbhb) (*Xie et al., 2016*) and isobutyrylation (Kibu) (*Zhu et al., 2021*), have been identified, indicating complex connections between cellular metabolism and epigenetic regulation of gene expression. Of note, while most fatty acylations were initially identified on nuclear histones, it is later found that reversible acylation serves as a significant regulatory mechanism for a diverse range of cellular proteins across multiple cellular compartments (*Park et al., 2013*). Indeed, growing evidence suggests a strong correlation between short-chain lysine acylations and diverse pathophysiological conditions, contributing to the progression of diseases (*Zhang et al., 2019*; *Pougovkina et al., 2014*; *Chen et al., 2020*).

Ketone bodies, including acetoacetate (AcAc), D-β-hydroxybutyrate (BHB), and acetone, are produced in the liver and subsequently distributed throughout the body during extended periods of dietary carbohydrate restriction and fasting. Apart from serving as an energy source, ketone bodies recently are starting to be acknowledged for their roles as signaling mediators, modulators of inflammation, and oxidative stress (*Puchalska and Crawford, 2017*). Their importance extends to critical pathological conditions that span cardiovascular, cerebrovascular, and neurological disorders such as failing hearts (*Bedi et al., 2016*), ischemic stroke (*Rahman et al., 2014*), Parkinson's disease, Alzheimer's disease (*Kashiwaya et al., 2000*; *Lim et al., 2011*), and cancers (*Shukla et al., 2014*; *Kang et al., 2015*). The Zhao group first found that β-hydroxybutyrate acts as a precursor for covalent modification of histones in the form of lysine β-hydroxybutyrylation (Kbhb) (*Xie et al., 2016*). In the mice liver subjected to fasting or streptozotocin-induced diabetes, Kbhb sets up a connection between chromatin regulation and the cellular pathophysiological functions of β-hydroxybutyrate (*Xie et al., 2016*). Further studies investigated key regulatory elements and substrate specificity of Kbhb (*Huang et al., 2021*). The β-hydroxybutyrate-mediated Kbhb of p53 pinpoints the connection between Kbhb and tumorigenesis (*Liu et al., 2019*). A recent study highlights that acetoacetate, another ketone molecule, also acts as a precursor for an uncharacterized histone PTM mark, known as lysine acetoacetylation (Kacac) (*Gao et al., 2023*). Nevertheless, key regulatory elements controlling Kacac formation, downstream targets of this modification at the proteomic level, as well as possible functional impacts of lysine acetoacetylation on cellular processes, remain as mysterious subjects of investigation for the field. These problems are particularly urgent to be addressed in order to gain a clear understanding of the regulatory mechanism of acetoacetate, especially considering its distinct function compared to β-hydroxybutyrate.

As a new PTM biomarker, there lacks effective technical tools to study Kacac in proteins. In recent years, the combination of specific antibody-based immunoprecipitation with advanced high-resolution mass spectrometry (MS) has streamlined the identification of PTM marks on proteins (*Sabari et al., 2017*). Nevertheless, the methods for generating new antibodies targeting small antigen markers such as protein acylation are typically laborious, time-intensive, expensive, and of low success rate. In this study, we introduce a distinctive and dependable chemo-immunological approach, wherein Kacac is first reduced into the Kbhb marker using sodium borohydride (NaBH$_4$) and subsequently detected using a currently commercially available anti-Kbhb antibody. This innovative method circumvents the necessity to develop new antibodies for Kacac, and meanwhile allows for simultaneous measurement and comparison of Kacac with Kbhb marks on proteins. Through implementing our developed chemo-immunological method, we present the systematic profiling of the Kacac mark as a novel, acetoacetate-induced modification in both histone and non-histone proteins in the human proteome. Our proteomic screen identified 139 unique Kacac sites across 85 proteins in HEK293T cells. Detailed bioinformatics analysis and RNA sequencing (RNA-seq) results reveal that Kacac has unique physiological significance and potentially participates in a balanced system with Kbhb to co-regulate cellular pathways. Expanding upon previously reported writers and erasers, we have identified p300, GCN5, and PCAF as acetoacetyltransferases in vitro, and HDAC3 as a de-acetoacetylase in vivo. Furthermore, we illustrated the mechanism behind acetoacetate-induced histone Kacac through enzymatic conversion by acetoacetyl-CoA synthetase (AACS). Together, this study delves into the complex mechanisms of ketone body-mediated Kacac, broadening the list of protein substrates and pathways potentially regulated by Kacac. The findings provide a foundational understanding for further exploration of the dynamic control mechanisms of

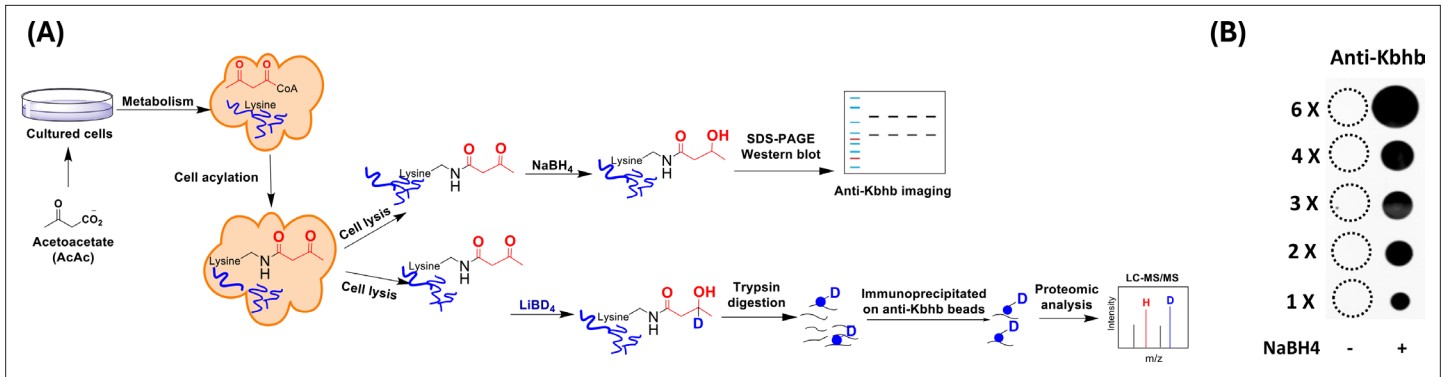

**Figure 1.** Methods development for the identification of lysine acetoacetylation. (**A**) Biosynthetic pathways for lysine acetoacetylation (Kacac) and proposed detection methods of Kacac. (**B**) Dot blot assay verifying the specificity of the pan anti-Kbhb antibody in our developed methods. Synthetic H2BK15acac peptide (Ac-PEPAKSAPAPKKGSKacacKAVTKAQKKDG-NH2) was used for the assay.

The online version of this article includes the following source data and figure supplement(s) for figure 1:

**Source data 1.** PDF file containing original dot blot for *Figure 1B*, indicating the relevant bands and treatments.

**Source data 2.** Original files for dot blot analysis displayed in *Figure 1B*.

**Figure supplement 1.** Scheme of chemical synthesis of the 2,5-dioxopyrrolidin-1-yl 2-(2-methyl-1,3-dioxolan-2-yl)acetate.

**Figure supplement 2.** Mass spectra (top) and ${}^1$H NMR spectrum (bottom) of ethyl 2-(2-methyl-1,3-dioxolan-2-yl)acetate.

**Figure supplement 3.** Mass spectra (top) and ${}^1$H NMR spectrum (bottom) of 2-(2-methyl-1,3-dioxolan-2-yl)acetic acid.

**Figure supplement 4.** Mass spectra (top) and ${}^1$H NMR spectrum (bottom) of 2,5-dioxopyrrolidin-1-yl 2-(2-methyl-1,3-dioxolan-2-yl)acetate.

**Figure supplement 5.** Structure (top) and ESI mass spectra (bottom) of the synthetic K15acac-H2B(1–26) peptide with the deconvoluted spectra below it.

ketone body homeostasis and the potential downstream roles of ketone bodies in regulating human disease processes.

## Results

### Method development for the detection of lysine acetoacetylation

Lysine acetoacetylation is a novel PTM, and currently there is no commercial antibody to study Kacac. To streamline the process of specific detection of Kacac for either immunoblotting or immunoprecipitation applications, we attempted to develop a reliable chemo-immunological approach for the detection of Kacac. Specifically, following the introduction of lysine acetoacetylation on proteins, the 3-carbonyl group in the Kacac marker was converted to a hydroxyl group using the reducing reagent NaBH$_4$ under slightly basic conditions. This conversion transforms the Kacac mark into β-hydroxybutyryllysine (Kbhb), which can then be detected using commercially available and widely used anti-Kbhb antibody. A big advantage of this chemo-immunological approach is that it enables the simultaneous detection of Kbhb and Kacac under various experimental conditions, and further allows for the identification and distinction of the Kacac mark from Kbhb through proteomic analysis if deuterated precursors are used (*Figure 1A*). To assess the practicability of our proposed method, we synthesized a peptide containing H2B N-terminal sequence with an acetoacetyl group at Lys-15 position, that is, H2B(1–26)K15acac (*Figure 1—figure supplements 1–5*). This is a Kacac site identified from our preliminary data. The acetoacetylated peptide was reduced using NaBH$_4$ under basic conditions, with a parallel sample lacking NaBH$_4$ under the same conditions serving as the control. The specificity of the pan anti-Kbhb antibody was then assessed via a dot blot assay (*Figure 1B*). In this assay, the pan-anti-Kbhb antibody demonstrated an excellent performance for the recognition of the Kacac peptide only after NaBH$_4$ reduction, supporting the feasibility of our developed chemo-immunological method. We want to point out that the pan-Kbhb antibody used herein is independent of peptide sequence so different modification sites on proteins shall all be recognized. Therefore, we subsequently employed this technical strategy to delve into the biology of lysine acetoacetylation.

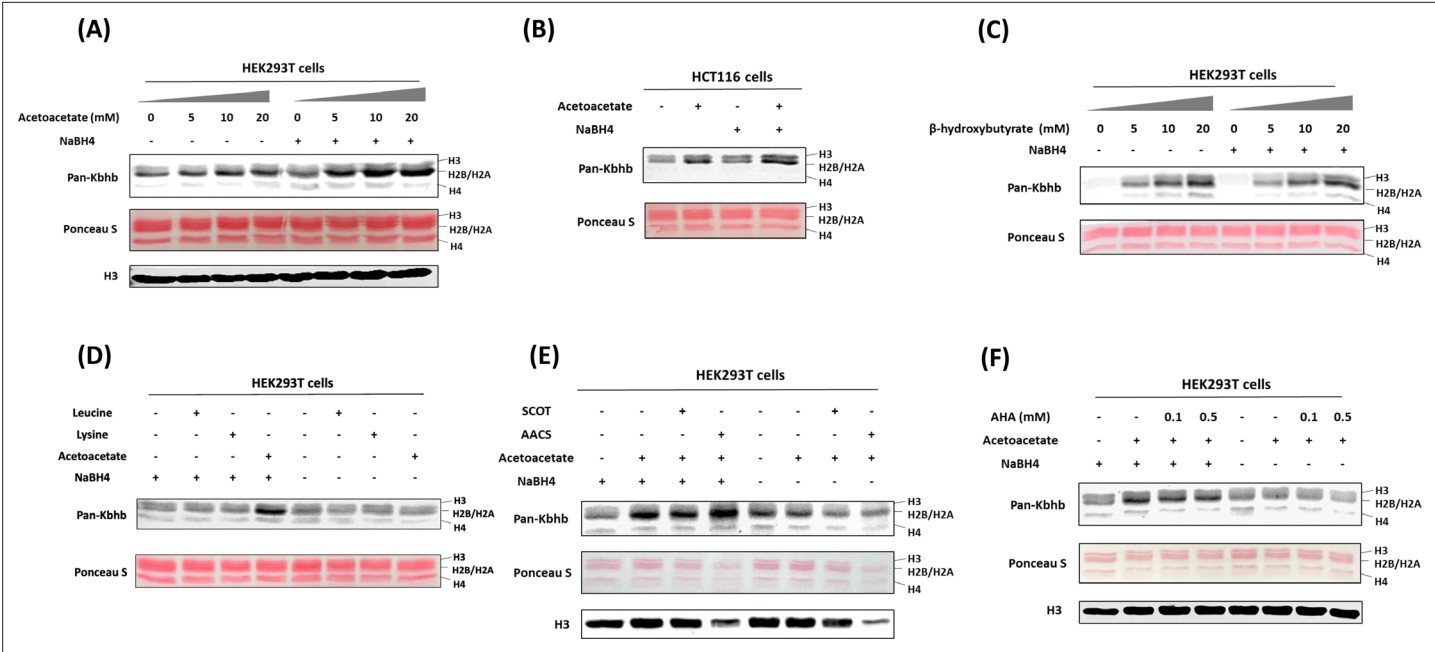

**Figure 2.** Acetoacetate dynamically regulates Kacac levels through the generation of acetoacetyl-CoA. (**A**) Western blot analysis of histones from HEK293T cells treated with increasing doses of lithium acetoacetate. (**B**) Detection of histone Kacac in HCT116 cells. (**C**) Western blot analysis of histones from HEK293T cells treated with increasing doses of sodium β-hydroxybutyrate. (**D**) Western blot analysis of histone Kacac in response to treatment with lithium acetoacetate or ketogenic amino acids (leucine and lysine) in HEK293T cells. (**E**) Western blot analysis of histone Kacac in response to overexpression of ketolysis enzymes (SCOT and AACS) in HEK293T cells. (**F**) Western blot analysis of histone Kacac in response to treatment with acetohydroxamic acid (AHA), a known SCOT inhibitor, in HEK293T cells.

The online version of this article includes the following source data and figure supplement(s) for figure 2:

**Source data 1.** PDF file containing original western blots for *Figure 2*, indicating the relevant bands and treatments.

**Source data 2.** Original files for western blot analysis displayed in *Figure 2*.

**Figure supplement 1.** Dynamic regulation of histone Kacac in vivo.

**Figure supplement 1—source data 1.** PDF file containing original western blots for *Figure 2—figure supplement 1*, indicating the relevant bands and treatments.

**Figure supplement 1—source data 2.** Original files for western blot analysis displayed in *Figure 2—figure supplement 1*.

## Acetoacetate is a precursor source for histone Kacac generation in cells

Recent studies have unveiled that short-chain fatty acids prompt the production of their respective acyl-CoAs within cells, consequently enhancing levels of histone acylations (*Fu et al., 2023*). We posit that acetoacetate enters the cell and is converted into acetoacetyl-CoA to serve as the cofactor for generating Kacac marks in cellular proteins. The recent study by Zhao et al. incubated HepG2 cells with ethyl acetoacetate and reported a dose-dependent increase in histone Kacac upon treatment, as detected by a specifically designed antibody (not commercially available) (*Gao et al., 2023*). To directly prove that acetoacetate induces Kacac levels, we cultured HEK293T cells with the treatment of lithium acetoacetate at different concentrations (0, 5, 10, and 20 mM). Following treatment, we extracted the nuclear histone proteins, which were then reduced using NaBH$_4$. The proteins were separated on a 15% sodium dodecyl sulfate–polyacrylamide gel electrophoresis (SDS–PAGE) gel, subsequently transferred to a nitrocellulose (NC) membrane, and probed using an anti-Kbhb antibody. Western blotting of histones revealed faint band intensities in the absence of NaBH$_4$ treatment, indicating the presence of endogenous Kbhb levels on the histones (*Figure 2A*). However, after reduction with NaBH$_4$, the intensity of the western blot bands increased substantially in an acetoacetate dose-dependent manner on the core histones. Clearly, this augmentation in western blot intensity was attributed to the heightened levels of Kacac resulting from acetoacetate treatment. Similarly, we observed an elevation in global histone Kacac levels in HCT116 cells, further supporting that acetoacetate served as the precursor for acetoacetylation (*Figure 2B*).

We also examined the time course of acetoacetate-induced Kacac and found that histone Kacac became detectable as early as 1 hr following acetoacetate treatment (*Figure 2—figure supplement 1A*). Notably, this chemo-immunological method has the advantage of allowing for the concurrent assessment of endogenous Kbhb levels and acetoacetate-induced Kacac levels on histones (*Figure 2A, B*). Our results indicated a subtle increase in histone Kbhb following acetoacetate treatment, suggesting that acetoacetate may play a minor role in promoting Kbhb generation within the cell.

To gain more insights into the dynamics of histone Kacac stimulated by acetoacetate, we treated HEK293T cells with other metabolic molecules that may influence Kacac levels within cells. We first investigated whether β-hydroxybutyrate could elevate histone Kacac levels in cells. By comparing the band intensities before and after $NaBH_4$ treatment, no increase in Kacac levels was observed following β-hydroxybutyrate treatment of HEK293T cells (*Figure 2C*). This suggests that β-hydroxybutyrate did not affect Kacac levels in HEK293T cells. To delve deeper into the precursor sources for histone Kacac generation, we examined the impact of two ketogenic amino acids—leucine and lysine—on Kacac levels. As shown in *Figure 2D*, only acetoacetate treatment caused Kacac increase; neither leucine nor lysine had an ability to increase Kacac levels on histones. These results indicated that acetoacetate acts as a predominant metabolic precursor for histone Kacac in the tested cells.

## AACS enzyme dynamically regulates histone Kacac in cells

It is known that AACS specifically catalyzes the activation of acetoacetate to its coenzyme A ester and serves as a highly regulated lipogenic enzyme in numerous tissues involved in de novo lipid synthesis (*Bergstrom, 2023*). We conjecture that exogenous acetoacetate elevates histone Kacac levels through directly generating acetoacetyl-CoA by the AACS. Thus, we investigated whether overexpression of the AACS would enhance histone Kacac levels in cells. We transiently transfected HEK293T cells with flag-tagged AACS plasmid followed by acetoacetate treatment. Immunoblot analysis of total cell lysates proved that AACS was successfully overexpressed at the protein level (*Figure 2—figure supplement 1B*). As anticipated, a substantial increase in histone Kacac levels was observed in HEK293T cells upon simultaneous overexpression of AACS and acetoacetate treatment (*Figure 2E*), supporting that AACS utilized acetoacetate to generate acetoacetyl-CoA priming for histone acetoacetylation. Succinyl CoA-oxoacid transferase (SCOT) is a mitochondrial enzyme thought to catalyze the regeneration of acetoacetyl-CoA from acetoacetate and facilitate ketone body oxidation for energy production (*Ma et al., 2024*). To further assess the potential contribution of SCOT to the acetoacetyl-CoA pool for histone Kacac, we conducted transient transfection to overexpress SCOT in HEK293T cells (*Figure 2—figure supplement 1B*). Nevertheless, our results did not reveal any significant increase in histone Kacac levels following SCOT overexpression (*Figure 2E*). We subsequently examined the effect of a previously described SCOT inhibitor, acetohydroxamic acid (AHA) (*Pickart and Jencks, 1979*), to validate the impact of SCOT on histone Kacac levels in HEK293T cells. Upon treating cells with AHA, we observed a slight decrease in signals, indicating a subtle effect of SCOT on histone Kacac (*Figure 2F*). Drawing from these findings, we conclude that AACS is the primary enzyme contributing to the acetoacetyl-CoA pool for histone Kacac generation in HEK293T cells, whereas SCOT exhibits minimal effects.

In a prior study, it was reported that AACS messenger RNA (mRNA) exhibited high expression levels in the kidney, heart, and brain, while demonstrating lower expression in the liver (*Ohgami et al., 2003*). However, AACS was strongly upregulated in liver cancer cells, influencing cancer development and progression (*Zhao et al., 2022*). The varied expression and distinct functions of AACS promoted us to investigate whether it exerts similar effects on histone Kacac levels in HepG2 liver cells. To accomplish this, we transiently overexpressed AACS in HepG2 cells (*Figure 2—figure supplement 1C*) and evaluated its impact on Kacac levels using our established methods. Surprisingly, the overexpression of AACS in HepG2 cells did not result in a significant increase in histone Kacac levels following acetoacetate treatment (*Figure 2—figure supplement 1D*). We also tested if HMG-CoA reductase (HMGCR) affects Kacac formation. We either overexpressed HMGCR or inhibited its activity using the lovastatin inhibitor in HepG2 cells. Subsequently, we treated the cells with acetoacetate to observe the impact of HMGCR on histone Kacac level. The chemo-immunological western blot results showed HMGCR modulation has little effect on the Kacac level (*Figure 2—figure supplement 1D, E*).

# p300, PCAF, and GCN5 mediate histone acetoacetylation

Mounting evidence suggests that classical lysine acetyltransferases can govern a range of newly discovered acyl modifications (*Sabari et al., 2017*). To comprehensively identify enzymes involved in mediating Kacac on histones, we sought to express and purify the major human histone acetyltransferases (HATs), including GCN5, PCAF, p300, Tip60, MOF, MOZ, HAT1, MORF, and HBO1. The SDS–PAGE analysis validated both the identity and the purity of these enzymes (*Figure 3—figure supplement 1A*). With these recombinant proteins, we tested their activity to acetylate or aceto-acetylate recombinant histone proteins through western blot assay. Individual HAT enzymes were incubated with histone H3/H4 and acetoacetyl-CoA in the reaction buffer. The reaction product was

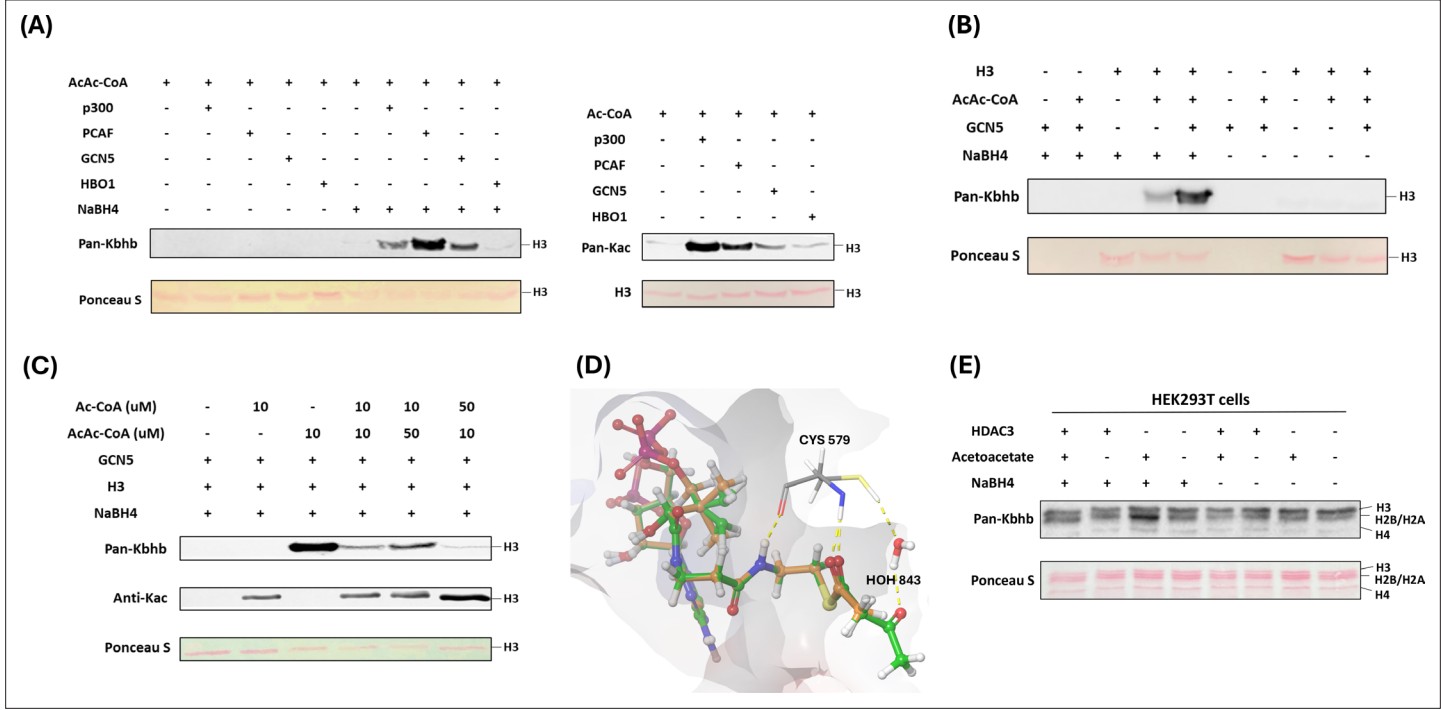

**Figure 3.** Identification of writers and erasers responsible for regulating histone Kacac. (**A**) p300, PCAF, and GCN5 exhibited remarkable acetoacetyltransferase activities (left) and acetyltransferase activities (right) on recombinant histone H3 proteins. (**B**) Validation of GCN5-mediated Kacac on recombinant histone H3 proteins. (**C**) Proportional changes of acyl-CoAs result in dynamics of substrates on recombinant histone H3 proteins. (**D**) Diagram illustrating the catalytic pocket of GCN5 bound with acetyl-CoA (orange) and acetoacetyl-CoA (green). PDB: 5TRL was used for the modeling. (**E**) Overexpression of HDAC3 abolished acetoacetate-induced Kacac in HEK293T cells.

The online version of this article includes the following source data and figure supplement(s) for figure 3:

**Source data 1.** PDF file containing original western blots for *Figure 3*, indicating the relevant bands and treatments.

**Source data 2.** Original files for western blot analysis displayed in *Figure 3*.

**Figure supplement 1.** Identification of writers responsible for regulating Kacac in vitro.

**Figure supplement 1—source data 1.** PDF file containing original gel or western blots for *Figure 3—figure supplement 1*, indicating the relevant bands and treatments.

**Figure supplement 1—source data 2.** Original files for sodium dodecyl sulfate–polyacrylamide gel electrophoresis (SDS–PAGE) or western blot analysis displayed in *Figure 3—figure supplement 1*.

**Figure supplement 2.** p300, GCN5, and PCAF act as acetoacetyltransferases in vitro.

**Figure supplement 2—source data 1.** PDF file containing original western blots for *Figure 3—figure supplement 2*, indicating the relevant bands and treatments.

**Figure supplement 2—source data 2.** Original files for western blot analysis displayed in *Figure 3—figure supplement 2*.

**Figure supplement 3.** Predicted binding modes of GCN5 with acyl-CoAs.

**Figure supplement 4.** Predicted binding modes of PCAF with acyl-CoAs.

**Figure supplement 5.** Predicted binding modes of p300 with acyl-CoAs.

**Figure supplement 6.** Predicted binding modes of HDAC3 with its substrate mimics.

detected using anti-Kbhb antibody, with and without NaBH$_4$ reduction. While most of these enzymes displayed well-characterized acetyltransferase activities, only p300, PCAF, and GCN5 demonstrated strong histone acetoacetyltransferase activities (*Figure 3A, B*, *Figure 3—figure supplement 1B–E*). Interestingly, GCN5 and PCAF exhibited a markedly higher acetoacetyltransferase-to-acetyltransferase ratio compared to that of p300. HAT1 and some MYST members exhibited weak acetoacetyltransferase activity (*Figure 3A*, *Figure 3—figure supplement 1C, D, F*). Subsequently, we conducted additional tests to further ascertain whether the acetoacetyltransferase activities of p300, GCN5, and PCAF could be detected on cellular histone substrates. This involved incubating acetoacetyl-CoA and histone extracts from HEK293T cells with p300, GCN5, or PCAF. As expected, stronger Kacac signals were observed in the groups with added enzymes (*Figure 3—figure supplement 1F*). To investigate whether p300 regulates histone Kacac in cells, we transiently transfected HEK293T cells with p300 plasmid, followed by treating the cells with acetoacetate. In accordance with the aforementioned observations (*Figure 2*), acetoacetate markedly elevated histone Kacac levels (*Figure 3—figure supplement 1G*). Unfortunately, the coexistence of p300 overexpression with acetoacetate treatment did not appreciably enhance Kacac levels (*Figure 3—figure supplement 1G*). It could be possible that native p300 protein levels in HEK293T cells are quite high, which shields the effect of the exogenously expressed proteins.

Next, we incubated recombinant p300, PCAF, or GCN5 with acetoacetyl-CoA and synthetic histone peptides under identical HAT assay conditions. The resulting reaction mixture was subjected to MALDI-MS analysis, which revealed expected product peaks (M+84) in all the catalytic reactions, thus providing further confirmation of the acetoacetyltransferase activities of the tested enzymes (*Figure 3—figure supplement 2A*).

Given that p300, GCN5, and PCAF can catalyze both histone Kac and Kacac, we predicted that alterations in a modified substrate would be directly influenced by fluctuations in the concentrations of acetyl-CoA or acetoacetyl-CoA within cells. These, in turn, would be closely linked to the regulation of these cofactors originating from both inside and outside the cell (*Sabari et al., 2015*). To gain insights into how acylations are differentially regulated in mammalian cells, we performed competition experiments by incubating recombinant histone H3 substrate with the HAT enzymes we tested above and varying ratios of acyl-CoAs. Indeed, altering the relative concentrations of acetyl-CoA or acetoacetyl-CoA showed differential impacts on the proportion of each modification present in the final reaction product. Acetyl-CoA was a superior cosubstrate of p300, GCN5, and PCAF. The presence of acetoacetyl-CoA did not affect acetyl-CoA-dependent histone acetylation (*Figure 3C*, *Figure 3—figure supplement 2B, C*). However, H3 acetoacetylation was significantly suppressed by the presence of acetyl-CoA, whereas increasing concentrations of acetoacetyl-CoA led to elevated Kacac levels. p300 was previously reported to catalyze the enzymatic addition of β-hydroxybutyryl group to lysine (*Huang et al., 2021*). To directly compare p300's activity as a β-hydroxybutyryltransferase or acetoacetyltransferase, we conducted additional incubations using recombinant human histone proteins along with varying ratios of acetoacetyl-CoA and β-hydroxybutyryl-CoA. Surprisingly, the presence of acetoacetyl-CoA scarcely influenced the generation of β-hydroxybutyrylation, suggesting that p300 exhibits stronger activity as a β-hydroxybutyryltransferase than an acetoacetyltransferase (*Figure 3—figure supplement 2D*).

To further understand the acetoacetylation activity of HATs described above, we conducted in silico molecular modeling of acetoacetyl-CoA/acetyl-CoA with HATs to explore potential binding interactions between acetoacetyl-CoA and the individual HATs. In the structure of GCN5 (PDB ID: 5TRL), the co-crystallized ligand is succinyl-CoA (*Wang et al., 2017*) and the carbonyl group of the succinate proved useful in validating the positioning of the acyl groups in the docked acetoacetyl-CoA and acetyl-CoA binding. Several residues were found to interact with both succinyl-CoA and the docked acetoacetyl-CoA and acetyl-CoA: Cys579, Val581, Val587, Gly589, Gly591, Thr592, Lys624, and Tyr617. The hydrogen bonding between the backbone of Ala614 and the co-crystallized succinyl-CoA is lost upon docking of acetyl-CoA and acetoacetyl-CoA. However, the catalytic water in the binding pocket retains its interactions with both acyl-CoAs. We observed that the distance between the sulfur of the co-crystallized succinyl-CoA and the thiol of Cys579 was maintained in the docking poses for the two acyl-CoAs with a distance of 6.3 Å for all three CoA derivatives (*Figure 3D*, *Figure 3—figure supplement 3*). In PCAF (PDB ID: 4NSQ), the co-crystallized ligand was CoA (*Shi et al., 2014*); many of the interactions found with this molecule were shared with the acyl-CoAs such as Cys574, Val576,

Val582, Gly584, Thr587, Lys619, and Tyr612. Acetoacetyl-CoA was found to have more steric clashes toward the terminal acetyl group, whereas acetyl-CoA seems to have a favorable hydrogen bonding interaction with Asp610. Here, the sulfurs of the CoA derivatives were found at the following distances from the catalytic Cys574: 3.43 Å for CoA, 3.59 Å for acetyl-CoA, and 3.34 Å for acetoacetyl-CoA. The western blot data suggest that acetyl-CoA may be a better substrate for PCAF (*Figure 3—figure supplement 2B*). This might be explained by the orientation of the acetyl group displayed in the modeling when compared to the acetoacetyl group. The carbonyl in the acetyl-CoA is in a position more similar to the Bürgi–Dunitz angle with respect to the thiol of Cys574, making the carbonyl more available for nucleophilic attack, whereas acetoacetyl-CoA has the carbonyl reversed with its oxygen pointing toward the thiol (*Figure 3—figure supplement 4*). For p300 (PDB ID: 5LKU), compared to the co-crystallized CoA in the structure (*Kaczmarska et al., 2017*), the interactions with Leu1398, Ser1400, Thr1411, Trp1466, and Ile1457 are maintained when docked with acetyl-CoA and aceto-acetyl-CoA. With acetyl-CoA, the pi–cation interaction with Arg1462, observed in the co-crystallized CoA, is maintained. However, the larger acyl group in acetoacetyl-CoA causes a slight shift in the adenyl ring, resulting in the loss of this interaction. Most importantly, the 3′ phosphate in the acyl-CoAs is shown to interact with Arg1410, which is typical for CoA binding (*Carrico et al., 2023*). The catalytic Cys1438 displays a distance of 6.61 Å between the side chain and the sulfur of CoA as well as acetyl-CoA and 6.64 Å for acetoacetyl-CoA (*Figure 3—figure supplement 5*).

## HDAC3 is a lysine deacetoacetylase in cells

In recent years, certain histone deacetylases (HDACs) have been reported to display non-canonical enzymatic activities toward various protein acylations, including de-succinylation (*Du et al., 2011*), de-β-hydroxybutyrylation (*Huang et al., 2021*), and others.

In a recent screening of individual HDACs using synthetic Kacac peptides, it was discovered that HDAC3 demonstrated distinct de-acetoacetylation activity in vitro (*Gao et al., 2023*). To investigate whether HDAC3 exhibits bona fide de-acetoacetylation activity in vivo, we conducted HDAC3 over-expression experiments in HEK293T cells through transient transfection. Then, we examined histone Kacac levels using our developed chemo-immunological method. Intriguingly, the overexpression of HDAC3 in HEK293T cells counteracted the increase in Kacac induced by acetoacetate treatment (*Figure 3E*). These findings support that HDAC3 functions as a deacetoacetylase both in vitro and in vivo.

To rationalize the de-acetoacetylation activity of HDAC3, small-molecule substrate mimics containing acyllysine (acetyllysine mimic or acetoacetyllysine mimic) were docked into the crystal structure of HDAC3 (*Figure 3—figure supplement 6*; *Watson et al., 2012*). In these molecules, lysine was alpha-N acetylated and C-amidated, so they more closely resemble the backbone of a protein rather than that of a standalone amino acid. The proposed binding mode positions the side chain nitrogen of acetyl lysine within 3.86 Å of zinc and 3.08 Å of Tyr298, while acetoacetyl lysine has its nitrogen within 4.50 Å of zinc and 4.24 Å of Tyr298. Additionally, both docked acyl lysines maintain hydrogen bonding with His134. The poses for both molecules coincide with the methionine model presented by *Watson et al., 2012* which can serve as a validation for the lysine substrate (*Watson et al., 2012*). When comparing the XP GScores, the acetoacetyl lysine mimic docking pose displayed a GScore of –8.899, while the acetyl lysine mimic had a GScore of –7.255. This may be because the docking model for the acetyl lysine mimic shows a steric clash between the methyl group of the acetyl and Cys145. In contrast, the acetoacetyl lysine mimic appears to bind further into the pocket, avoiding this steric interaction. This computational model provides a structural insight into the de-acetoacetyl-ation activity of HDAC3.

## Proteome-wide identification of Kacac substrates in human cells

To understand the scope and impact of lysine acetoacetylation, we mapped out the landscape of Kacac substrates in the HEK293T cell proteome. To do so, the cellular proteome was extracted and reduced using $LiBD_4$, followed by trypsin digestion. The resulting tryptic peptides were enriched using anti-Kbhb antibody-conjugated resin. The enriched peptides were then released from the resin for high-performance liquid chromatography (HPLC)–MS/MS analysis and database searching. We iden-tified 24 histone Kacac modification sites located at H1, H2A, H2A.Z, H2B, H3, and H4: 10 were the same as the previously reported and 14 were new sites (*Figure 4A*). Additionally, our methods enabled

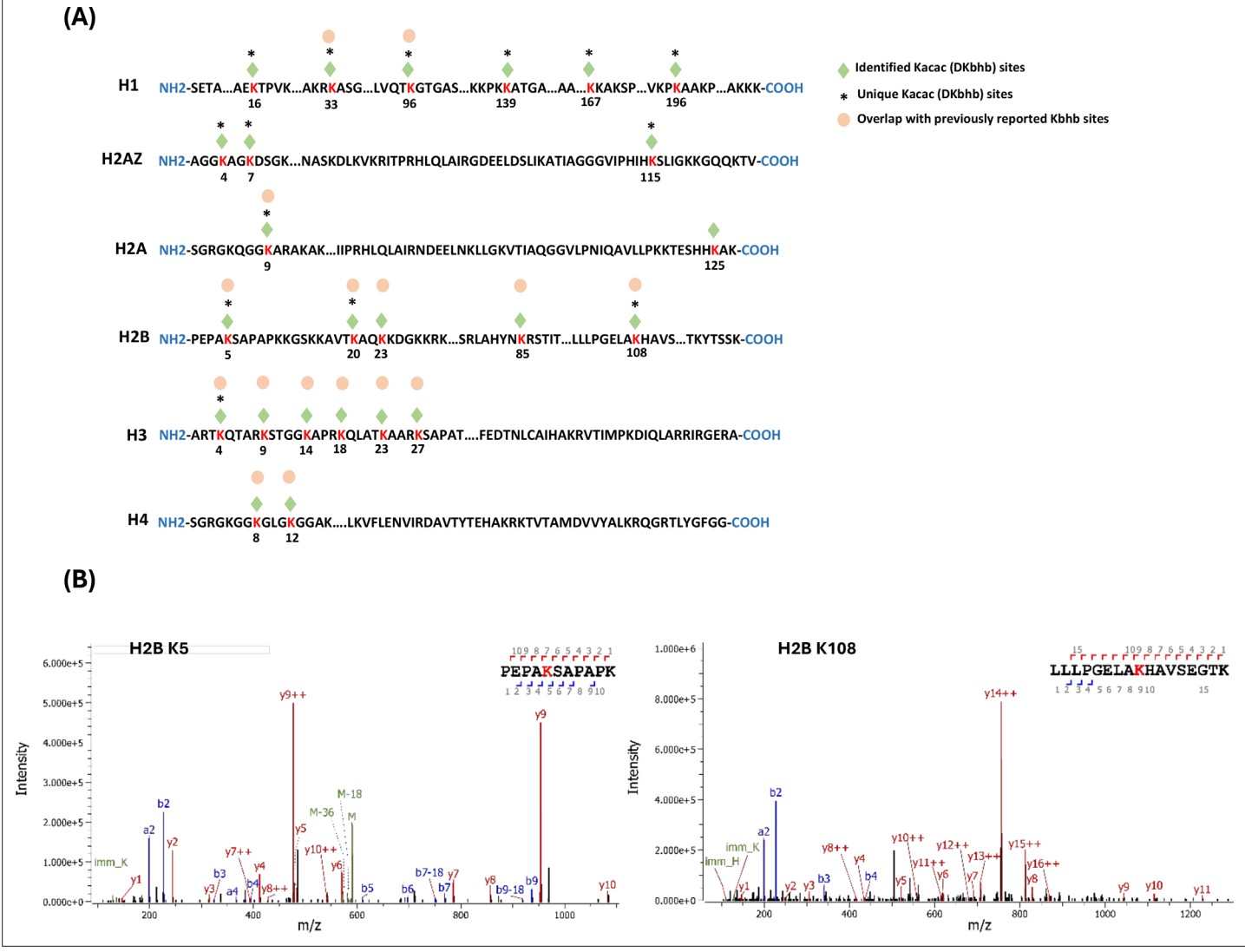

**Figure 4.** Proteomic screening of histone Kacac sites in HEK293T cells. (**A**) Illustration of histone DKbhb (Kacac) sites identified in HEK293T cells. Green diamond indicates Kacac sites detected in our study. * denotes previously unknown histone Kacac sites. For comparison, the overlapped known Kbhb sites (labeled with an orange dot) described in the literature are also listed. (**B**) MS/MS spectra of two representative DKbhb peptides derived from HEK293T histones.

the successful identification of 139 sites on 85 cellular proteins (*Figure 5A*, *Figure 5—source data 1*). Among these Kacac (DKbhb) proteins, 60 proteins (70%) contain a single Kacac site, 13 proteins (15%) contain two Kacac sites, and the rest contain three or more Kacac sites (*Figure 5A*). Impressively, nucleophosmin (NPM) contains 4 Kacac sites, nucleolar and coiled-body phosphoprotein 1 (NOLC1) protein and histone H2B protein bear 5 Kacac sites, histone H3 protein bears 6 sites, and nucleolin (NUCL) protein bears 10 sites. It is worth emphasizing that in this experiment, the Kacac marks were identified as the deuterated Kbhb (DKbhb) form. Importantly, endogenously existing Kbhb was also detected from this single experiment. Therefore, our chemo-immunological method, combined with MS/MS, enables the simultaneous identification of both Kacac and Kbhb sites. To examine potential Kacac motifs in the identified substrate proteins, we compared the amino acid sequences surrounding Kacac sites against the background of the human proteome. The data revealed a preference for Ser, Ala, or Gly at the −1 and +1 positions. Additionally, proline was highly enriched at the −2 positions, while Leu was largely depleted at most positions. Interestingly, a significant preference was observed for alanine and positively charged lysine at multiple positions (−6, −5, −4, 1, 2, 3, 4, 5, and 6), whereas negatively charged amino acids (Asp and Glu) were consistently underrepresented at the −1 and −6

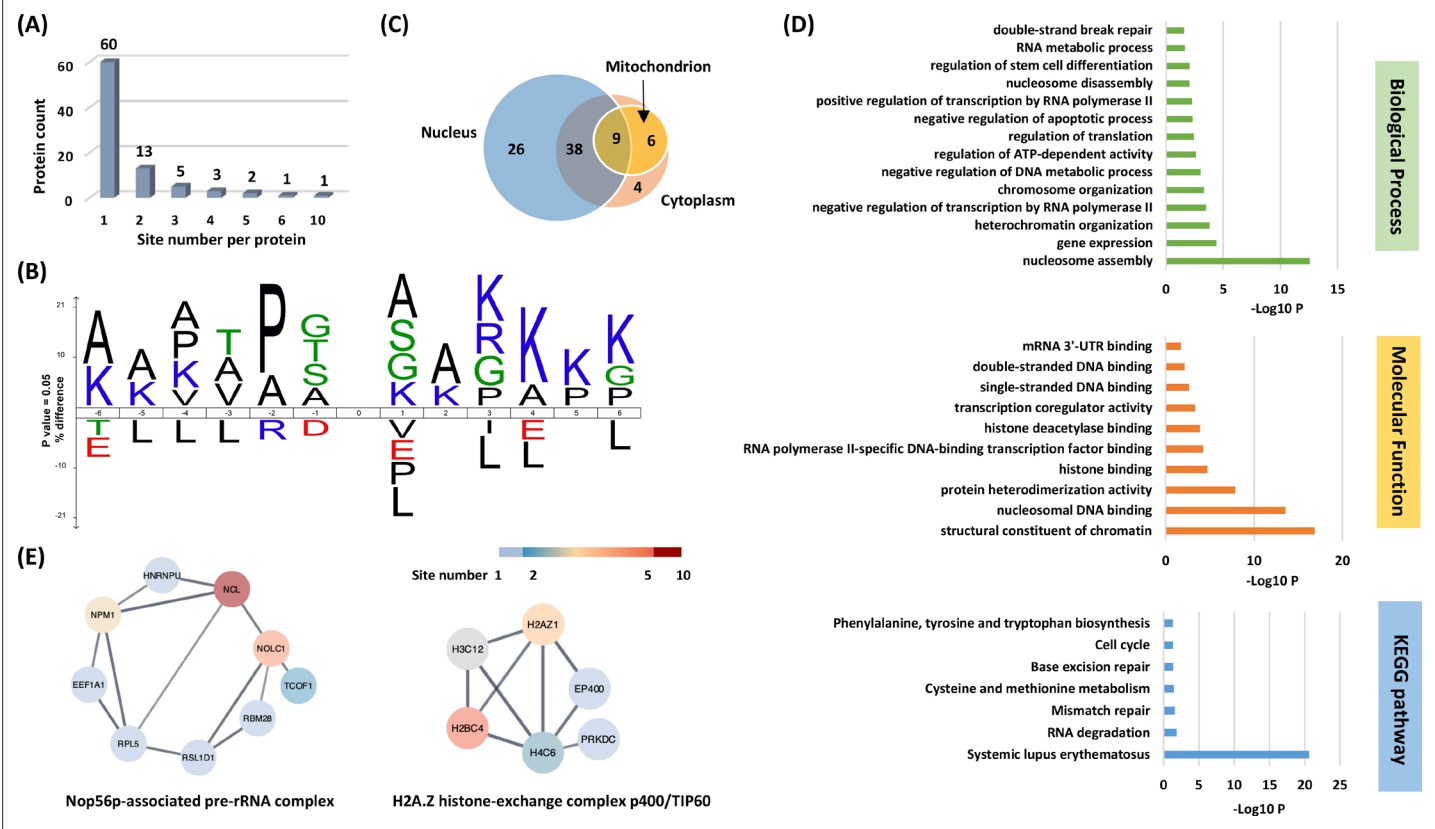

**Figure 5.** Systematic profiling of the Kacac proteome. (**A**) Distribution of the Kacac protein based on the site number per protein. (**B**) The consensus sequence logos show enrichment of amino acid residues among the Kacac sites. (**C**) Venn diagram shows the cellular compartment distribution of Kacac proteins. (**D**) Representative ontology annotations and all Kyoto Encyclopedia of Genes and Genomes (KEGG) pathways enriched within the Kacac proteome. (**E**) Two protein complexes significantly enriched in the Kacac proteome. The color bar depicts the number of Kacac sites identified in each protein.

The online version of this article includes the following source data and figure supplement(s) for figure 5:

**Source data 1.** Complete list of identified Kacac (DKbhb) sites.

**Source data 2.** Kacac sites (**A**) and neighbor sites (**B**) that are critical for biological functions.

**Source data 3.** Protein complex analysis of Kacac proteins.

**Figure supplement 1.** Functional annotation of Kacac marks in HEK293T cells.

positions (*Figure 5B*). These features closely align with the reported flanking sequence preference of Kbhb but differ from the motif analysis results of lysine acetylation (Kac), crotonylation (Kcr), malonylation (Kmal), succinylation (Ksucc), and 2-hydroxyisobutyrylation (Khib) (*Park et al., 2013*; *Svinkina et al., 2015*; *Huang et al., 2018c*; *Huang et al., 2018b*; *Huang et al., 2018a*; *Nishida et al., 2015*).

To investigate the subcellular distribution of Kacac substrates within cells, we conducted a cellular compartment analysis of the Kacac proteome. Previous studies have indicated that Ksucc and Kmal are notably enriched in mitochondria, while the subcellular localization of Kac, Khib, and Kbhb substrates is often observed in either the cytoplasm or the nucleus (*Park et al., 2013*; *Huang et al., 2021*; *Huang et al., 2018b*; *Huang et al., 2018a*; *Colak et al., 2015*). The subcellular distribution of Kacac closely resembles that of Kac, Khib, and Kbhb substrates, with only 17% Kacac proteins located in mitochondrion. Also, Kacac proteins are significantly more represented in the nucleus, accounting for 85% of all proteins (*Figure 5C*).

To discern the physiologically relevant Kacac sites, we compared our dataset with public databases of known PTMs in UniProt (http://www.uniprot.org). Through analysis, we discovered 73 Kacac sites overlapping with other modifications (*Figure 5—source data 2A*), encompassing ubiquitination and previously reported Kac, butyrylation (Kbu), Kcr, propionylation (Kpr), lactylation (Kla), Kbhb, Ksucc, Kmal, glutarylation (Kglu), Khib, as well as lysine methylation (Kme) sites. These PTMs may compete

for the same lysine residues and reciprocally influence one another. For example, p53 Kbhb competes with p53 Kac, leading to reduced cell growth arrest and apoptosis (*Liu et al., 2019*). In principle, Kacac could occupy one of the same residues targeted by Kbhb and Kac. Moreover, we conducted a comparison between the Kacac sites and reported mutations linked to protein biological functions in the UniProt database. The findings revealed seven lysine sites significantly impacted by mutations, leading to altered protein functions (*Figure 5—source data 2A*). Protein methylation and acetylation play pivotal roles in mediating crucial biological processes and signaling pathways. In our findings, several crucial proteins known as substrates for acetylation or methylation were discovered to be acetoacetylated, including p53, histone H2A.Z, and SMC3. Acetoacetylation at K370 of p53 may contribute to the regulation of p53's activation or repression, thereby influencing rapid responses to DNA damage or cellular stress (*Huang et al., 2006*). Similarly, Kacac at K4 of histone H2A.Z may impair GCN5-mediated H2A.Z.1 acetylation, thereby affecting RNA polymerase II recruitment and gene trans-activation (*Semer et al., 2019*). In addition to methylation and acetylation, Kacac might influence the interaction of regulatory proteins with their protein partners by interfering with other reversible PTMs, such as the small ubiquitin-like modifier (SUMO) conjugation pathway. One such example is the identification of acetoacetylation at K102 of SERBP1. However, SUMOylation of SERBP1 at K102, K228, and/or K281 by SUMO proteins is implicated in the regulation of $As_2O_3$-induced PML nuclear bodies (PML-NBs) formation, which is associated with oxidative stress and interferon stimulation (*Saito et al., 2017*). Furthermore, four lysine residues (K38KKK) located in the N-terminal domain of caspase-7 facilitate the rapid proteolysis of poly(ADP-ribose) polymerase 1 (PARP-1), ensuring swift cell demise during apoptosis (*Desroches and Denault, 2019*). Therefore, acetoacetylation of K38 (K38acac) may reduce the ability of caspase-7 to cleave PARP1. These findings suggest a potential interplay between Kacac and other PTMs within protein contexts, offering insights into the biological functions of Kacac and its role in complex metabolic regulation. In addition to competing with other modifications for the same residues, we identified 78 Kacac sites located within five residues of various types of PTMs, including lysine acylation, phosphorylation, *O*-GlcNAcylation, ubiquitination, and ADP-ribosylation. Notably, some of these Kacac sites are situated near critical mutation sites associated with protein dysfunction (*Figure 5—source data 2B*). Together, these results imply the widespread and profound biological impacts of Kacac by functionally influencing substrates across the proteome.

## Functional annotation of the lysine acetoacetylome

To elucidate the biological functions of Kacac substrates in mammalian cells, we conducted a gene ontology (GO) analysis. Our findings revealed significant enrichment of Lys-acetoacetylated proteins in cellular metabolic processes, particularly in nucleosome assembly ($p = 2.82 \times 10^{-13}$) and gene expression ($p = 3.98 \times 10^{-5}$). Nucleosome assembly and disassembly are essential aspects of chromatin dynamics, critically involved in DNA replication, transcription, and repair. During nucleosome assembly and disassembly, histone chaperones bind to histones to prevent their abnormal aggregation and improper interactions with DNA, thereby facilitating their proper transfer onto DNA chains to form nucleosomes (*Gurard-Levin et al., 2014*). Interestingly, we identified several histone chaperones that can undergo lysine acetoacetylation, including SET protein, NPM, and SSRP1, a component of the FACT (facilitates chromatin transcription) complex. In addition, ATP-dependent chromatin remodeling complexes modulate chromatin packing by sliding, ejecting, or restructuring nucleosomes, thereby enabling dynamic regulation of chromatin architecture (*He et al., 2020*). We also identified Kacac-modified substrates within ATP-dependent chromatin remodeling complexes, including BAZ1A from the ISWI complex and SMARCC2 from the SWI/SNF complex. These findings highlight the potential significant role of Kacac in nucleosome assembly. Of further interest, we found that the molecular functions attributed to Kacac substrates included structural constituent of chromatin ($p = 1.41 \times 10^{-17}$), nucleosomal DNA binding ($p = 2.75 \times 10^{-14}$), protein heterodimerization activity ($p = 1.26 \times 10^{-8}$), histone binding ($p = 1.92 \times 10^{-5}$), and RNA polymerase II-specific DNA-binding transcription factor binding ($p = 5.39 \times 10^{-5}$) (*Figure 5D*). The Kyoto Encyclopedia of Genes and Genomes (KEGG) pathway enrichment analysis indicated that Kacac proteins were enriched in immune response, RNA/DNA metabolism, amino acid metabolism, and cell cycle (*Figure 5D*). Notably, most Kacac proteins are involved in systemic lupus erythematosus ($p = 2.49 \times 10^{-21}$), RNA degradation ($p = 1.45 \times 10^{-2}$), and mismatch repair ($p = 2.59 \times 10^{-2}$). To characterize the classification of the Kacac proteome, we conducted a protein class enrichment analysis. Chromatin/chromatin-binding, or -regulatory protein,

HMG box transcription factor, and RNA metabolism protein were prominently represented (*Figure 5—figure supplement 1A*). Chromatin-binding or regulatory proteins and HMG box transcription factors play key roles in regulating chromatin architecture, gene expression, DNA repair, and other cellular processes. In addition to the proteins identified in these two chromatin-related classes, a substantial number of Kacac-modified proteins are associated with RNA processing. For example, splicing factor 3b subunit 1 (SF3B1) is a core component of the U2 snRNP at the spliceosome's catalytic center, where it is essential for recognizing and defining the 3′ splice site at intron–exon junctions (*Wahl et al., 2009*). Other proteins, including HNRNPL and HNRNPC from the heterogeneous nuclear ribonucleoprotein (hnRNP) family, as well as NONO (non-POU domain-containing octamer-binding protein), also play crucial roles in RNA splicing (*Geuens et al., 2016*; *Ronchetti et al., 2024*). Collectively, these findings suggest potential regulatory roles for Kacac in chromatin remodeling, gene expression, and RNA processing.

To identify protein complexes regulated by Kacac, we performed protein complex enrichment analysis with the manually curated CORUM database (*Ruepp et al., 2010*). Our analysis revealed significant enrichment of Kacac substrates in several protein complexes (*Figure 5—source data 3*), including the Nop56p-associated pre-rRNA complex ($p = 7.46 \times 10^{-9}$), the H2A.Z histone-exchange complex p400/TIP60 ($p = 7.98 \times 10^{-9}$), the LARC complex (LCR-associated remodeling complex) ($p = 6.01 \times 10^{-8}$), the TLE1 corepressor complex ($p = 6.97 \times 10^{-8}$), the MASH1 promoter–coactivator complex ($p = 1.25 \times 10^{-7}$), the SNF2h–cohesin–NuRD complex ($p = 1.73 \times 10^{-6}$), the PID complex ($p = 1.81 \times 10^{-6}$), and the Spliceosome, E complex ($p = 8.85 \times 10^{-5}$) (*Figure 5E*, *Figure 5—source data 3*). The Nop56p-associated pre-rRNA complex is linked to ribosome biogenesis in human cells (*Hayano et al., 2003*). The H2A.Z histone-exchange complex p400/TIP60 regulates H2A.Z deposition, thereby influencing chromatin structure and gene expression (*Obri et al., 2014*). Our data revealed that the highest number of Kacac-modified substrates belong to these two complexes, with acetoacetylation detected in 9 of 104 subunits of the Nop56p-associated pre-rRNA complex and 6 of 27 subunits of the H2A.Z histone-exchange complex p400/TIP60, respectively. Notably, several components of the Nop56p-associated pre-rRNA complex were extensively modified with Kacac, including NUCL, NPM, and NOLC1 (*Figure 5E*). These findings highlight the potential involvement of Kacac in protein complexes associated with chromatin remodeling, gene expression, and ribosome biogenesis. Additionally, the LARC exhibits a sequence-specific binding to the hypersensitive 2 (HS2)-Maf recognition element (MARE) DNA and orchestrates nucleosome remodeling (*Mahajan et al., 2005*). Spliceosomes catalyze the splicing of primary gene transcripts (pre-mRNA) to produce mRNA. Our findings reveal that 5 out of 19 subunits within the LARC complex and 6 out of 129 subunits in the Spliceosome E complex undergo acetoacetylation, affirming the significance of Kacac in nucleosome assembly and a correlation between Kacac and RNA processing. Moreover, the TLE1 corepressor complex is recognized for its involvement in HES1-mediated transcriptional repression (*Ju et al., 2004*). Intriguingly, our analysis revealed acetoacetylation in 4 of the 8 subunits of this complex, with NUCL presenting 10 Kacac sites and the crucial protein PARP1 showing 2 Kacac sites. The HDAC1-containing PID complex, which is involved in repressing p53 transactivation functions and contributes to the regulation of p53-mediated growth arrest and apoptosis (*Luo et al., 2000*), is also enriched in our study. Notably, our data revealed acetoacetylation in 3 out of 5 subunits within the PID complex, shedding light on the contribution of Kacac to cell growth and proliferation. Together, these findings suggest that Kacac participates in various cellular functions, including nucleosome remodeling, RNA metabolism, and transcriptional regulation.

## Profiling physiological relevance of Kacac modification

To explore the epigenetic role of Kacac in gene regulation, we performed RNA-seq from control and acetoacetate-treated HEK293T cells. We compared the differentially regulated genes between the two conditions and conducted GO analysis, KEGG pathway enrichment analysis, and gene set enrichment analysis (GSEA) to further understand the biological implications of Kacac. In our analysis, 4764 genes were found to be upregulated, whereas 2822 genes were downregulated after acetoacetate treatment (|fold change| >1.5, adjusted p value <0.05) (*Figure 6A*, *Figure 6—source data 1*). We extracted differentially expressed genes (DEGs) between the 'control' and 'AcAc' conditions and generated a heatmap illustrating the expression patterns of the DEGs across both conditions (*Figure 5—figure supplement 1B*). GO analyses revealed that genes primarily enriched in downregulated terms are

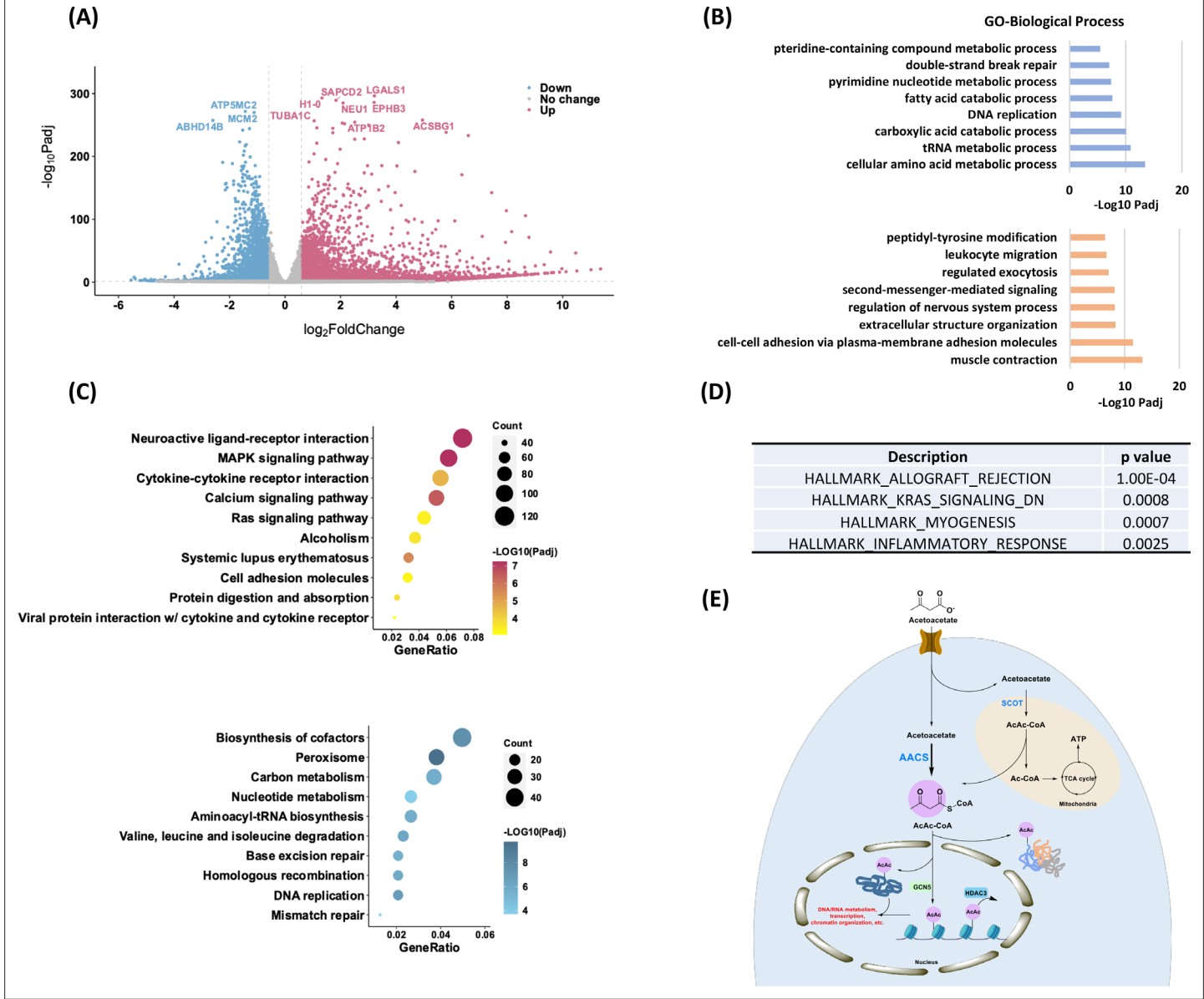

**Figure 6.** Profiling of physiological relevance of Kacac mark in HEK293T cells. (**A**) Volcano plot analysis of pairwise comparison of RNA sequencing (RNA-seq) results from HEK293T cells with or without 20 mM acetoacetate treatment. (**B**) Gene ontology (GO, biological process) enrichment analysis of downregulated (blue) and upregulated (orange) differentially expressed genes (DEGs) after lithium acetoacetate treatment in HEK293T cells, ranked on the basis of adjusted p values. (**C**) Bubble plots showing the top 10 Kyoto Encyclopedia of Genes and Genomes (KEGG) pathways enriched in the upregulated (upper) and downregulated (bottom) DEGs after lithium acetoacetate treatment in HEK293T cells, ranked on the basis of adjusted p values and counts. Gradient colors represent enriched significance, and size of circles represents numbers of DEGs. (**D**) Hallmark gene sets identified by gene set enrichment analysis (GSEA) after lithium acetoacetate treatment in HEK293T cells, ranked on the basis of p values. (**E**) A graphical model of Kacac. In this model, AACS, not SCOT, is a major player for AcAc-CoA and Kacac generation from acetoacetate.

The online version of this article includes the following source data for figure 6:

**Source data 1.** Differentially expressed genes (DEGs) upon acetoacetate treatment in RNA sequencing (RNA-seq).

**Source data 2.** Gene ontology (GO) terms identified by using differentially expressed genes (DEGs).

**Source data 3.** Kyoto Encyclopedia of Genes and Genomes (KEGG) pathways identified by using differentially expressed genes (DEGs).

associated with metabolic processes such as amino acid metabolic processes, carboxylic acid catabolic processes, fatty acid catabolic processes, and pteridine-containing compound metabolic processes (*Figure 6B*, *Figure 6—source data 2A*). Additionally, DNA/RNA-associated processes such as tRNA metabolic processes, pyrimidine nucleotide metabolic processes, DNA replication, and DNA repair

are prominently represented among downregulated terms (*Figure 6B*). Several other processes, such as mitochondrial gene expression, translation, cell cycle checkpoint signaling, and protein localization to peroxisomes, are substantially downregulated following acetoacetate treatment (*Figure 6—source data 2A*). In contrast, upregulated DEGs are predominantly enriched in processes including muscle contraction, cell adhesion, cellular component organization, regulation of nervous system processes, second-messenger-mediated signaling, regulated exocytosis, and leukocyte migration (*Figure 6B*). Moreover, peptidyl-tyrosine phosphorylation, regulation of blood circulation, heart processes, and the phospholipase C-activating G-protein-coupled receptor signaling pathway exhibit notably elevated expression following lithium acetoacetate treatment (*Figure 6—source data 2B*). In line with the GO analysis findings, subsequent KEGG pathway enrichment analysis revealed that downregulated DEGs induced by acetoacetate are significantly enriched in pathways related to peroxisome function, cofactor biosynthesis, DNA replication and repair, amino acids, and carbon metabolism (*Figure 6C*, *Figure 6—source data 3*A). Conversely, upregulated DEGs are predominantly associated with pathways such as neuroactive ligand–receptor interaction, the MAPK signaling pathway, the calcium signaling pathway, systemic lupus erythematosus, cytokine–cytokine receptor interaction, among others (*Figure 6C*, *Figure 6—source data 3*B). Additional GSEA results revealed enrichment of genes related to allograft rejection, KRAS signaling, muscle differentiation, and inflammatory response (*Figure 6D*). These findings further support that acetoacetate-mediated Kacac is closely linked to cellular processes involving amino acid metabolism, DNA/RNA metabolism, gene expression, translation, proliferation, and immune response.

## Discussion

Intracellular short-chain fatty acids and their corresponding acyl-CoAs play a pivotal role in regulating short-chain lysine acylations on cellular proteins, establishing a link between cellular metabolism and gene expression (*Sabari et al., 2017*; *Fu et al., 2023*). Ketone bodies are crucial metabolites in maintaining physiological homeostasis and vital metabolic and signaling mediators in diverse physiological and pathological states (*Puchalska and Crawford, 2017*). Recent studies have reported that β-hydroxybutyrate serves as a precursor for Kbhb on both histones and non-histones, significantly expanding its role as a protein function modifier (*Xie et al., 2016*; *Huang et al., 2021*). The Kbhb levels are dynamic in response to altered physiological conditions such as starvation and diabetic ketoacidosis (*Xie et al., 2016*).

Despite the recent disclosure unveiling histone Kacac as a novel PTM (*Gao et al., 2023*), the mechanisms underlying the dynamics and proteome-wide distribution for Kacac marks are largely unexplored. A thorough elaboration of the regulatory elements governing Kacac substrate proteins, along with the enzymes responsible for adding and removing this modification, is crucial for the functional characterization of Kacac and understanding the role of acetoacetate in cellular regulation. In this study, we developed a chemo-immunological approach for Kacac detection and applied it to conduct the first global proteomic analysis of the lysine acetoacetylome in human cells. The approach not only offers a reliable and time-efficient method for detecting Kacac without the need to develop new antibodies but also importantly enables the simultaneous detection and quantification of Kbhb and Kacac in the same biological samples. In our proteomic profiling, we identified 139 lysine acetoacetylated peptides across 85 proteins in HEK293T cells. We found that Kacac extensively modifies numerous biologically important proteins, notably at critical sites, indicating its significant roles in diverse cellular processes and disease development. Notably, we discovered 14 previously unidentified histone Kacac marks such as H3K4, H2BK5, H2BK20, H2BK108, H2AK9, and several marks on H1 and H2A.Z. Many of the identified Kacac sites overlapped with, or were in close proximity to, other PTMs, including acylation, methylation, phosphorylation, O-GlcNAcylation, ubiquitination, and ADP-ribosylation. The crosstalk between Kacac and other PTMs may influence protein localization, interactions with binding partners, or other functions, ultimately leading to altered biological outcomes. For example, acetoacetylation of NOLC1 at K579—located within the CK2α-binding region (residues 568–596) and near Ser574, which significantly enhances CK2α-binding affinity—may influence phosphorylation at Ser574 and NOLC1's interaction with CK2α, potentially altering CK2's phosphorylation activity and its downstream signaling events (*Lee et al., 2013*). Furthermore, we found that Kacac proteins exhibited a notable enrichment in the nucleus. In line with this finding, Kacac substrates were concentrated in proteins associated with various nuclear biological processes and signaling pathways,

including chromatin organization, DNA repair, RNA metabolism, as well as gene expression. Moreover, a considerable fraction of Kacac proteins was identified in the cytosol, while only a minor proportion was observed in mitochondria, mirroring the subcellular distribution patterns of Kac, Khib, and Kbhb substrates (*Huang et al., 2021*; *Huang et al., 2018b*; *Huang et al., 2018a*). However, this distribution differs from that of Ksucc or Kmal substrate proteins (*Park et al., 2013*; *Colak et al., 2015*), which are predominantly localized in mitochondria, indicating the dynamic functions of different lysine acylations.

Structurally different lysine acylations correlate with distinct physiological states and may lead to functional consequences. Sequence preference analysis revealed an enrichment of both negatively charged amino acids and positively charged lysine, with proline being largely depleted at most positions related to the Kcr or Khib sites (*Huang et al., 2018c*; *Huang et al., 2018a*). Near the Ksucc sites, glycine, alanine, and lysine were preferentially located, whereas arginine and serine were predominantly depleted at most positions (*Park et al., 2013*). While sharing most sequence preference similarities with the reported Kbhb (*Huang et al., 2021*), the flanking sequence analysis of Kacac substrates revealed a higher enrichment of alanine at more positions near Kacac sites. Functional analysis further substantiates the unique and distinct regulatory roles of various acylations. For instance, Kac predominantly targets proteins involved in RNA biology, while Ksucc and Khib are notably enriched in multiple metabolism-related pathways (*Park et al., 2013*; *Huang et al., 2018b*). As the two primary ketone bodies, acetoacetate-mediated Kacac does indeed exhibit similarities to β-hydroxybutyrate-mediated Kbhb in terms of functional annotation, both of which are associated with DNA repair, RNA metabolism, and chromatin organization. However, Kbhb and Kacac still play distinct roles in specific processes. For instance, most Kbhb-modified proteins are involved in spliceosome, ribosome function, and RNA transport, whereas Kacac-modified proteins are enriched in systemic lupus erythematosus and RNA degradation pathways. Compared to previously reported acylated substrates, 43 out of 85 Kacac proteins are preferred substrates of various acylation marks, including Kbz, Khib, Kac, and Kbhb (*Huang et al., 2021*; *Huang et al., 2018b*; *Huang et al., 2018a*; *Tan et al., 2022*). These overlapping proteins participate in key cellular processes such as chromatin remodeling, gene expression, apoptosis, and DNA/RNA metabolism—including DNA repair, RNA biosynthesis, and RNA splicing—highlighting the shared functional roles of protein lysine acylations. Among the 139 Kacac sites, 69, 44, 47, and 84 overlapped with Kbz, Kac, Khib, and Kbhb sites, respectively, with Kbhb showing the highest degree of overlap with Kacac compared to the other PTMs. Additionally, comparison of the Kacac proteome with the reported Kbhb-modified proteins reveals that 75 out of 85 proteins carry both modifications, indicating significant overlap in substrate specificity between these two ketone body-mediated PTMs. Although different PTM markers can occupy the same lysine sites on overlapping substrate proteins, their functional consequences might be distinct. For example, PARP1 is activated by DNA damage and catalyzes the ADP-ribosylation of multiple proteins, leading to the recruitment of DNA repair factors and chromatin remodeling (*Ray Chaudhuri and Nussenzweig, 2017*). Interestingly, lactylation of PARP1 at seven lysine sites (K498, K505, K506, K508, K518, K521, and K524) enhances its ADP-ribosylation activity, whereas acetylation reduces it, indicating that lysine acylations differentially regulate PARP1 activity and may influence the DNA damage response (*Sun et al., 2022*). Furthermore, some Kacac sites coincide with residues previously reported to be modified by other lysine acylations implicated in disease progression. For example, malonylation of NUCL at K124 and K398 triggers its translocation and promotes AKT translation, thereby driving cell proliferation and tumor growth in hepatocellular carcinoma (*Sun et al., 2024*). Acetylated NPM at K229 and K230 functions as a coactivator during RNA polymerase II-driven transcription and regulates the expression of genes that promote oral tumorigenesis (*Senapati et al., 2022*). Another study reported that lactylation of chromobox 3 (CBX3) at K10 promotes its interaction with H3K9me3 and facilitates gastrointestinal cancer progression (*Duan et al., 2024*). These findings suggest a potential interplay between Kacac and other lysine acylation marks at protein sites, as well as a pathological significance of Kacac, although its precise functions remain to be elucidated.

In a prior study, p300 was identified as a β-hydroxybutyryltransferase, while HDAC1 and HDAC2 were characterized as de-β-hydroxybutyrylases (*Huang et al., 2021*). In our biochemical screening, we demonstrated that p300 and GCN5 function as acetoacetyltransferases. This finding is consistent with a previous study that highlighted the modest activity of p300 and substantial activity of GCN5 in transferring the acetoacetyl motif onto histone proteins (*Gao et al., 2023*). p300 has been

shown to catalyze a wide range of acylations—including propionylation, butyrylation, crotonylation, 2-hydroxyisobutyrylation, succinylation, and β-hydroxybutyrylation—owing to its unique acyl-CoA-binding pocket (*Sabari et al., 2017*; *Sabari et al., 2015*; *Chen et al., 2007*; *Liu et al., 2017*; *Kaczmarska et al., 2017*). The absence of such a pocket results in the limited activity of GCN5, which efficiently catalyzes propionylation but is poorly active for butyrylation and crotonylation (*Leemhuis et al., 2008*; *Ringel and Wolberger, 2016*). Surprisingly, GCN5 appears to exhibit relatively higher acetoacetyltransferase activity compared to p300; however, further evidence is needed to clarify their catalytic mechanisms as acetoacetyltransferases. We found that PCAF also exhibits robust lysine acetoacetyltransferase activity. Given that PCAF possesses conserved functional domains and exhibits high sequence similarity with GCN5 (75% amino acid identity) (*Koutelou et al., 2021*), it is not surprising that our data highlight its notable activities in vitro. HBO1 is a versatile protein acyltransferase that catalyzes acetylation, propionylation, butyrylation, crotonylation, lactylation, benzoylation, and acetoacetylation (*Gao et al., 2023*; *Tan et al., 2022*; *Xiao et al., 2021*; *Niu et al., 2024*). A previous study elucidated that the CoA moiety, rather than the acyl group of acyl-CoAs, plays a major role in binding to HBO1 (*Xiao et al., 2021*). Unfortunately, our test did not demonstrate the acetoacetyltransferase activity of HBO1 as reported in a previous study (*Gao et al., 2023*). This may be attributed to the low activity of the recombinant HBO1 used in our assay. HBO1 has been reported to assemble into a multisubunit complex that includes ING4/5, hEaf6, and scaffold proteins from either the JADE1/2/3 or BRPF1/2/3 families, which strongly enhance its acyltransferase activity (*Xiao et al., 2021*). Further investigation of the enzymatic kinetics of p300, PCAF, GCN5, and possibly other HATs, both in isolation and within their complexes, would provide valuable insights.

PCAF and GCN5 are involved in key processes through their roles as acetyltransferases. PCAF, for instance, acetylates p53, influencing the protein's DNA-binding ability and apoptotic activities following DNA damage (*Liu et al., 1999*; *Chao et al., 2006*). PCAF also induces acetylation of HMGA1 proteins in PC-3 human prostate cancer cells and acetylates HMGB1, which in turn regulates the release of HMGB1 and is linked to the cellular inflammatory response (*Zhang et al., 2007*; *Hwang et al., 2014*). Interestingly, we found that several substrates previously identified as acetylation targets of GCN5/PCAF, including p53 and high mobility group (HMG) proteins, can also undergo acetoacetylation. Using recombinant histone proteins as substrates, we demonstrated that acetoacetyl-CoA can compete with acetyl-CoA for binding to PCAF/GCN5. Therefore, it is of great interest to further investigate how GCN5/PCAF regulate dynamic PTMs, including Kacac, and their effects across various metabolic processes or pathological conditions. Beyond establishing p300, GCN5, and PCAF as Kacac writers, we demonstrate that HDAC3 acts as an eraser of Kacac in cellulo. Further research is needed to investigate the unique biological effects of HDAC3-mediated de-Kacac and to identify additional HDACs that possess de-Kacac activity under physiological conditions.

Short-chain acyl-CoAs, derived from their respective short-chain fatty acids, act as cofactors for acyltransferases, facilitating various lysine acylation reactions (*Xie et al., 2016*; *Fu et al., 2023*). Our data demonstrate that acetoacetate, rather than the catabolism of ketogenic amino acids leucine and lysine, primarily contributes to Kacac formation in histones. This observation is consistent with previous research showing that ketone bodies primarily originate from the oxidation of fatty acids, with leucine contributing only up to 4% of the carbon in ketone bodies (*Puchalska and Crawford, 2017*; *Puchalska and Crawford, 2021*). Herein, we investigated the mechanism by which acetoacetate generates acetoacetyl-CoA for histone Kacac. We explored the contributions of AACS and SCOT in utilizing acetoacetate to increase the acetoacetyl-CoA pool for histone Kacac. Interestingly, we found that AACS demonstrates a substantial capacity, while SCOT exhibits a lower ability to convert exogenous acetoacetate for histone Kacac in HEK293T cells. Endogenous SCOT levels in HEK293T cells may be saturated for Kacac generation, explaining why SCOT overexpression did not significantly impact Kacac levels in our study, while SCOT inhibition did. These findings shed light on the mechanism by which AACS utilizes acetoacetate to generate acetoacetyl-CoA in cytosol, subsequently transferring it into the nucleus for histone Kacac. This notion is in line with an earlier report suggesting that nuclear acyl-CoAs can equilibrate with the cytosolic acyl-CoAs pool across nuclear pores, facilitating histone acylation and thereby influencing gene expression regulation (*Sabari et al., 2017*; *Naquet et al., 2020*). There is evidence supporting the existence of significant pools of CoA and acyl-CoAs in mitochondria and peroxisomes (*Horie et al., 1981*; *Van Broekhoven et al., 1981*). However, it remains unclear whether and how acyl-CoAs synthesized in these compartments can contribute to the

nuclear acyl-CoA pool for histone acylations. As a crucial mitochondrial ketolytic enzyme, SCOT may predominantly contribute to Kacac on mitochondrial proteins rather than nuclear histones. A recent study revealed that SCOT is widely distributed throughout the cell and serves as a lysine succinyl-transferase for global lysine succinylation (Ksucc) on proteins in mammalian cells (**Ma et al., 2024**). Notably, SCOT mediates Ksucc on mitochondrial protein LACTB, leading to enhanced mitochondrial membrane potential and respiration in liver cancer cells (**Ma et al., 2024**). Therefore, it is also possible that SCOT is widely distributed throughout the cell and has additional roles in acetoacetate utilization beyond its function as a ketolytic enzyme in the mitochondria.

AACS expression greatly increased histone Kacac levels in HEK293T cells. In contrast, AACS did not show significant effects in increasing histone Kacac levels in HepG2 cells. It is possible that even if histone Kacac is regulated by AACS in HepG2 cells, its turnover rate is slow, or there may be limited availability of free CoA in the HepG2 cells, which could hinder the further generation of acetoace-tyl-CoA after AACS overexpression. It is also possible that AACS regulation in HepG2 cells is more complex, and its activity is dynamically influenced by several factors. Given the low Km of AACS for acetoacetate and the consistent presence of acetoacetate at concentrations sufficient to saturate or nearly saturate AACS, the utilization of acetoacetate as an anabolic substrate is governed not by the level of acetoacetate production, but rather by the regulation of AACS expression and activity. The mRNA level or activity of hepatic AACS is intricately regulated by factors such as modulators of cholesterol synthesis, acyl-CoAs, and physiological states like fasting and obesity (**Bergstrom, 2023**). Furthermore, the utilization of acetoacetate for lipid synthesis is significantly augmented in highly malignant hepatoma cells, as evidenced by its pathway remaining unaffected by hydroxycitrate inhibi-tion (**Hildebrandt et al., 1995**). These findings support our hypothesis that AACS is intricately regu-lated by various factors in HepG2 cells, although additional evidence is needed to substantiate this claim. Ketone body metabolism through AACS plays a pivotal role in maintaining cholesterol homeo-stasis (**Hasegawa et al., 2012**). Therefore, we further investigated the contributions of HMGCR, a rate-limiting enzyme in the cholesterol biosynthetic pathway, to histone Kacac levels. However, neither the overexpression of HMGCR nor the blockade of cholesterol biosynthesis by lovastatin significantly influenced histone Kacac levels. This suggests that HMGCR-associated cholesterol biosynthesis does not compete for acetoacetate production, thus not affecting Kacac in HepG2 cells.

The ratio of acetoacetate to β-hydroxybutyrate is directly linked to the mitochondrial $NAD^+$/NADH ratio, and the equilibrium constant of β-hydroxybutyrate dehydrogenase 1 (BDH1) favors the produc-tion of β-hydroxybutyrate (**Krebs et al., 1969**; **Williamson et al., 1967**). A cytoplasmic β-hydroxy-butyrate dehydrogenase (BDH2), sharing 20% sequence identity with BDH1, possesses a high Km for ketone bodies and facilitates the exclusive $NAD^+$-dependent conversion of cytosolic β-hydroxy-butyrate to acetoacetate (**Guo et al., 2006**; **Davuluri et al., 2016**). This enzyme serves either as a secondary system for energy supply or as a contributor to generate precursors for lipid and sterol synthesis (**Guo et al., 2006**). BDH1 likely primarily facilitates the conversion between acetoacetate and β-hydroxybutyrate, which may explain our findings that acetoacetate induces Kbhb to some extent, while β-hydroxybutyrate treatment promotes Kbhb rather than Kacac. However, the extent of BDH1/BDH2 involvement and their regulatory role in the equilibration of acetoacetate and β-hydroxy-butyrate for their corresponding PTMs requires further investigation.

The RNA-seq analyses revealed that acetoacetate extensively influences gene expression, partic-ularly in processes/pathways involved in amino acid and carbon metabolism, DNA/RNA metabolism, DNA transcription, nervous system regulation, immune response, proliferation, and inflammatory response. These enriched processes/pathways closely overlap with those identified in our proteomic data, affirming the biological effects of acetoacetate-mediated Kacac. When comparing our RNA-seq data with previous data on Kbhb, we identified several pathways predominantly enriched in aceto-acetate, rather than β-hydroxybutyrate treatment. These pathways include biosynthesis of cofactors, homologous recombination, the MAPK signaling pathway, alcoholism, and hypertrophic cardiomyop-athy, indicating a distinctive regulatory pattern of acetoacetate. Interestingly, we noticed an inverse trend in the signaling pathways induced by acetoacetate compared to those induced by β-hydroxy-butyrate. For instance, pathways such as peroxisome, valine, leucine, and isoleucine degradation, ribosome, and RNA degradation were suppressed following acetoacetate treatment, while they were upregulated in response to an increase in β-hydroxybutyrate levels (**Xie et al., 2016**). Conversely, pathways upregulated by acetoacetate are prominently enriched in neuroactive ligand–receptor

interaction, the calcium signaling pathway, systemic lupus erythematosus, cytokine–cytokine receptor interaction, and cell adhesion molecules, while these pathways are significantly downregulated in response to increased β-hydroxybutyrate levels (*Xie et al., 2016*). Similarly, disease-related pathways, including systemic lupus erythematosus and type I diabetes mellitus, are upregulated by acetoacetate while significantly downregulated in response to β-hydroxybutyrate stimulation (*Xie et al., 2016*). Previous studies have shown evidence supporting that acetoacetate may possess an opposite repertoire of signaling functions compared to β-hydroxybutyrate in certain aspects. For instance, acetoacetate activates GPR43 and sustains energy homeostasis by regulating lipid metabolism under ketogenic conditions (*Miyamoto et al., 2019*). However, β-hydroxybutyrate exerts an antilipolytic effect by activating the GPR109A receptor on adipocytes, thereby creating a negative feedback loop where ketosis suppresses lipolysis in adipocytes (*Taggart et al., 2005*). Furthermore, high concentrations of acetoacetate may trigger a pro-inflammatory response, whereas β-hydroxybutyrate exerts a predominantly anti-inflammatory response (*Puchalska and Crawford, 2017*). Similarly, the administration of exogenous β-hydroxybutyrate-induced hepatic fibrosis, while acetoacetate inhibited it (*Puchalska et al., 2019*). Collectively, these findings, along with our data, support the notion that acetoacetate may possess distinct functions and potentially form a balanced system with β-hydroxybutyrate to co-regulate cellular processes, partially achieved through Kacac and Kbhb.

With the discovery of Kacac as a new PTM in the proteome, many open questions surge to the biological and biomedical fields. The effectiveness of ketogenic diets in certain diseases is partly attributed to the SCOT-mediated terminal oxidation of acetoacetate. Evidence indicates a reduction in SCOT expression under diabetic conditions and during myocardial injury (*Hasan et al., 2010*; *Wang et al., 2021*). Inborn errors in SCOT function or SCOT knockout lead to severe consequences such as ketoacidosis, lethargy, and even neonatal death (*Fukao et al., 2004*; *Cotter et al., 2011*). In addition to their role in mitochondrial function, ketone bodies may contribute to disease-related metabolic regulation by participating in de novo lipid synthesis. AACS, the key enzyme in ketone body utilization for lipid synthesis, is upregulated and implicated in the development and progression of hepatocellular carcinoma (*Zhao et al., 2022*). Acetoacetate, rather than β-hydroxybutyrate, can be readily utilized via AACS and is the preferred substrate for lipid synthesis compared to glucose and acetate (*Bergstrom et al., 1984*). Given the pivotal role of SCOT/AACS in critical regulatory effects and their potential mediation of Kacac, it is imperative to extensively explore the SCOT/AACS-mediated Kacac substrates to elucidate the mechanisms underlying acetoacetate-associated diseases. Although acetoacetate and β-hydroxybutyrate, as two main metabolic ketone bodies, are well recognized as catabolic substances, our proteomic data and RNA-seq data support that acetoacetate-mediated Kacac functions differently from β-hydroxybutyrate-mediated Kbhb and may form a balance cycle with Kbhb in certain pathways, adapting to (patho)physiological changes. Notably, acetoacetate displayed unique roles in cancer cells, including the promotion of tumor growth in BRAF-V600E+ cancer cells through the activation of MEK–ERK signaling and induction of FGF21 expression in HepG2 cells (*Puchalska and Crawford, 2021*). All these studies underscore the importance of extensively investigating the unique regulatory functions of acetoacetate. Hence, further detailed functional studies of Kacac substrates in specific disease models can be conducted to gain a deeper understanding of the biological mechanisms of acetoacetate. Of further note, some protein readers (e.g., bromodomain, YEATS domain, and PHD domain) of PTM marks display varying binding affinities toward different PTM marks (*Sabari et al., 2017*). Therefore, comprehending the distinct reader proteins that specifically recognize Kacac is crucial for investigating the diverse biological processes of Kacac that set it apart from other types of PTMs.

In summary, we innovated an enabling chemo-immunological approach for Kacac protein detection and enrichment, in combination with high-resolution tandem mass spectrometer and functional analyses, to unveil acetoacetate-mediated Kacac as a pivotal molecular mechanism for the global modification of cellular proteins and the regulation of cellular metabolism. We discovered several regulatory elements of lysine acetoacetylation, expounding its dynamics in cells. Our study not only opens a new window for investigating the crosstalk between metabolism and epigenetic regulation but also lays a foundation for further exploring the diverse roles of Kacac in moderating environmental

impacts on metabolism and diseases related to ketone bodies. Importantly, our study uncovers an additional avenue through the dynamic coordination of Kacac and Kbhb pathways in response to metabolic changes, contributing to the maintenance of cellular homeostasis.

# Materials and methods

**Key resources table**

| Reagent type (species) or resource | Designation | Source or reference | Identifiers | Additional information |
|---|---|---|---|---|
| Cell line (Human) | HEK293T | Gift | N/A | Further STR profiling showed a close match (~80%) to HEK293-derived cell lines (e.g., 293TT), consistent with the known genetic similarity and instability among HEK293 derivatives; no mycoplasma contamination was found. |
| Cell line (Human) | HCT116 | Gift | N/A | Further STR profiling identified the cells as HCT116 (90% match); no mycoplasma contamination was found. |
| Cell line (Human) | HepG2 | ATCC | Cat#: HB-8065, RRID:CVCL_0027 | Further STR profiling identified the cells as HepG2 (100% match); no mycoplasma contamination was found. |
| Transfected construct (Human) | pCMVβ-p300-myc | Addgene | Cat#: 30489 | |
| Transfected construct (Human) | flag-HDAC3 | Addgene | Cat#: 13819 | |
| Transfected construct (Human) | AACS | OriGene Technologies | Cat#: RC206247 | |
| Transfected construct (Human) | SCOT | OriGene Technologies | Cat#: RC203764 | |
| Transfected construct (Human) | HMGCR | Addgene | Cat#: 86085 | |
| Antibody | Pan anti-Kbhb (Rabbit polyclonal) | PTM BioLabs | Cat#: PTM-1201, RRID:AB_2927634 | WB (1:1000) |
| Antibody | Pan anti-Kac (Rabbit polyclonal) | PTM BioLabs | Cat#: PTM-105, RRID:AB_2877698 | WB (1:2000) |
| Antibody | Anti-Flag (Mouse monoclonal) | Thermo Fisher Scientific | Cat#: MA1-91878, RRID:AB_1957945 | WB (1:1000) |
| Antibody | Anti-β-actin (Mouse monoclonal) | Santa Cruz Biotechnology | Cat#: sc-47778, RRID:AB_626632 | WB (1:1000) |
| Antibody | Anti-H3 (Mouse monoclonal) | Santa Cruz Biotechnology | Cat#: sc-517576, RRID:AB_2848194 | WB (1:3000) |
| Antibody | Anti-rabbit IgG HRP-linked antibody | Cell Signaling Technology | Cat#: 7074S, RRID:AB_2099233 | WB (1:3000) |
| Antibody | Anti-mouse IgG HRP-linked antibody | Cytek Biosciences | Cat#: 72-8042, RRID:AB_3750732 | WB (1:3000) |
| Peptide, recombinant protein | Histone H3.1 Human | New England Biolabs | Cat#: M2503 | |
| Peptide, recombinant protein | Histone H4 Human | New England Biolabs | Cat#: M2504 | |
| Peptide, recombinant protein | H3(1–20)/H4(1–20) | This paper | N/A | Described in in vitro HAT activity screening |
| Peptide, recombinant protein | K15acac-H2B(1–26) | This paper | N/A | Described in dot blot assay |
| Commercial assay or kit | ECL Western Blotting Substrate | Thermo Fisher Scientific | Cat#: 32209 | |

*Continued on next page*

*Continued*

| Reagent type (species) or resource | Designation | Source or reference | Identifiers | Additional information |
|---|---|---|---|---|
| Commercial assay or kit | M-PER Mammalian Protein Extraction Reagent | Thermo Fisher Scientific | Cat#: 78501 | |
| Commercial assay or kit | EpiQuik Total Histone Extraction Kit | Epigentek | Cat#: OP-0006-100 | |
| Commercial assay or kit | RNeasy Plus Mini Kit | QIAGEN | Cat#: 74134 | |
| Commercial assay or kit | Lipofectamine 3000 Transfection Reagent | Thermo Fisher Scientific | Cat#: L3000008 | |

## Common chemicals

Unless otherwise noted, all chemical reagents were purchased from Sigma-Aldrich (St. Louis, MO). For synthesis of the peptide, the following Nα-Fmoc protected amino acids and reagents were purchased from ChemPep Inc (Wellington, FL): Fmoc-Ser(tBu)-OH, Fmoc-Gly-OH, Fmoc-Lys(Boc)-OH, Fmoc-Ala-OH, Fmoc-Thr(tBu)-OH, Fmoc-Gln(Trt)-OH, Fmoc-Pro-OH, Fmoc-Val-OH, Fmoc-Asp(tBu)-OH, Fmoc-Glu(tBu)-OH, Rink Amide resin, and 1-ethyl-3-(3-dimethylaminopropyl)carbodiimide hydrochloride (EDC HCl). Other chemicals used for peptide synthesis were *N*-methylpyrrolid-2-one, *p*-toluenesulfonic acid, and *N*-hydroxysuccinimide from Oakwood Chemical (Estill, SC); *O*-(1*H*-6-chlorobenzotriazole-1-yl)-1,1,3,3-tetramethyluronium hexafluorophosphate (HCTU) and sodium chloride from Chem-Impex (Dale, IL); ethane dithiol and ethyl acetoacetate from Alfa Aesar (Haverhill, MA); thioanisole and trifluoroacetic acid (TFA) from Acros Organics (Geel, Belgium); diisopropylethylamine (DIPEA) from TCI America (Portland, OR); dimethyl sulfoxide (DMSO) and diethyl ether from Fisher Scientific (Hampton, NH); ethylene glycol and sodium hydroxide from VWR (Radnor, PA); sodium bicarbonate from Avantor (Radnor, PA); hydrochloric acid from Aqua solutions (Jasper, GA).

## Chemical synthesis

Synthesis of K15acac-H2B(1–26) peptide

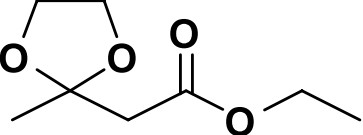

**Chemical structure 1.** Ethyl 2-(2-methyl-1,3-dioxolan-2-yl)acetate.

To a flask, ethyl acetoacetate (1.96 ml, 15.368 mmol, 1 equiv.), ethylene glycol (1.12 ml, 19.978 mmol, 1.3 equiv.), and *p*-toluenesulfonic acid (29.2 mg, 0.154 mmol, 0.01 equiv.) were added to 20 ml of benzene, and refluxed overnight using a Dean-Stark trap. After cooling to room temperature, benzene was removed by rotary evaporation and diluted in 10 ml ethyl acetate. The organic solvent was then washed with 30 ml 5% aqueous sodium bicarbonate, then 30 ml brine. The organic layer was dried over sodium sulfate before removal by rotary evaporation to obtain the product, ethyl 2-(2-methyl-1,3-dioxolan-2-yl)acetate as a colorless oil. Yield: 2.592 g, 96.82%. ESI-HRMS calc for $C_8H_{14}NaO_4$ [M+Na]⁺: 197.0784 found 197.0779 ¹H NMR (500 MHz, DMSO-*d₆*) δ 3.95 (q, *J* = 7.1 Hz, 2H), 3.77 (s, 4H), 1.29 (s, 3H), 1.08 (t, *J* = 7.1 Hz, 3H).

**Chemical structure 2.** 2-(2-Methyl-1,3-dioxolan-2-yl)acetic acid.

The ketal ester (***Palm and Thompson, 2017***) (2.592 g, 14.88 mmol, 1 equiv.) was dissolved in 20 ml of ethanol before adding a 2 M solution of sodium hydroxide (0.774 g, 9.67 ml, 19.344 mmol, 1.3 equiv.). The resulting solution was stirred at room temperature for 4 hr. The ethanol was removed by rotary evaporation, and the resulting aqueous phase was subsequently washed with 2 × 20 ml of diethyl ether. The aqueous phase was then acidified with 2 M HCl. The product was extracted from the aqueous phase with 3 × 100 ml of ethyl acetate. The organic phase was dried over sodium sulfate and then removed with rotary evaporation to obtain the product, 2-(2-methyl-1,3-dioxolan-2-yl)acetic acid as a yellow oil. 781 mg, 35.91%. ESI-HRMS calc for $C_6H_{10}NaO_4$ [M+Na]$^+$: 169.0471 found 169.0466 $^1$H NMR (500 MHz, DMSO-$d_6$) $\delta$ 3.84 (s, 4H), 3.15 (s, 2H), 1.37 (s, 3H).

**Chemical structure 3.** 2,5-Dioxopyrrolidin-1-yl 2-(2-methyl-1,3-dioxolan-2-yl)acetate.

The protected acetoacetate (***DeBerardinis and Thompson, 2012***) (598 mg, 4.09 mmol, 1 equiv.) was dissolved in 10 ml of dichloromethane before adding N-hydroxysuccinimide (753 mg, 6.54 mmol, 1.6 equiv.) and EDC HCl (1.25 g, 6.54 mmol, 1.6 equiv.) to the solution. The solution was stirred at room temperature for 3 hr. The solution was diluted with 100 ml of dichloromethane, and then the organic layer was washed with 50 ml of brine, 50 ml of saturated aqueous sodium bicarbonate, and finally 50 ml of brine. The organic layer was dried over sodium sulfate and subsequently evaporated to obtain the compound, 2,5-dioxopyrrolidin-1-yl 2-(2-methyl-1,3-dioxolan-2-yl)acetate as a yellow oil, with no further purification. 670 mg, 67.32%. ESI-HRMS calc for $C_{10}H_{13}NNaO_6$ [M+Na]$^+$: 266.0635 found 266.0628 $^1$H NMR (500 MHz, DMSO-$d_6$) $\delta$ 3.86–3.79 (m, 4H), 2.90 (s, 2H), 2.71 (s, 4H), 1.34 (s, 3H).

## Solid-phase peptide synthesis

The starting peptide (PEP AKS APA PKK GSK(Dde) KAV TKA QKK DG) was synthesized via standard Fmoc solid-phase peptide synthesis using the FOCUS XC automatic synthesizer (AAPPTec, Louisville, KY). Rink amide resin was swollen in 15 ml of N,N-dimethylformamide (DMF) for 15 min and drained thereafter. 8 ml of 20% (vol/vol) piperidine in DMF was added and left to deprotect Fmoc for 10 min. This was repeated with a washing step before and after the second addition of piperidine. 5 ml of the corresponding protected amino acid (0.2 M in DMF, 5 equiv.) was activated in a separate vessel using 5 ml of HCTU (0.2 M in DMF, 5 equiv.) and 5 ml of 4-methylmorpholine (NMM) (0.2 M in DMF, 5 equiv.). The solution was mixed for 1 min before being added to the resin. The reaction proceeded for 1 hr before proceeding to the draining and washing steps. The deprotection and coupling steps were repeated until the desired peptide sequence was acquired. Acetylation of the N-terminus was accomplished manually by adding a solution of 4:1 DMF to acetic anhydride (50 equiv.) and DIPEA (12.5 equiv.) to the resin. The reaction was allowed to mix for 30 min with subsequent draining and washing by DMF. The deprotection was accomplished manually as well by addition of 8 ml of 2% $N_2H_4$ in DMF. The reaction was mixed for 2 hr followed

by the draining and washing of the resin. This was repeated once before additional washing and drying of the resin.

### Coupling acetoacetate

The peptide on resin was swollen in 20 ml DMF for 30 min then drained. A solution of 5 ml of 2,5-dioxopyrrolidin-1-yl 2-(2-methyl-1,3-dioxolan-2-yl)acetate (0.2 M in DMF, 5 equiv.) and 5 ml of NMM (0.2 M in DMF, 5 equiv.) was added to the resin and mixed for 3 hr at room temperature. The resin was then drained, washed, and dried.

### Cleavage

Removal of the peptide from the resin was accomplished using 5 ml of 95% TFA, 2.5% $H_2O$, and 2.5% triisopropylsilane and allowed to mix for 4 hr at room temperature. The resulting solution was filtered to remove the resin and diluted with 40 ml of cold diethyl ether and centrifuged at 3214 × $g$. The solution was decanted from the precipitate and 40 ml of cold diethyl ether was added again and followed by a second centrifugation step. The solution was decanted, and the precipitate was dissolved in water for HPLC purification using LC-20AT HPLC and SPD-20A UV/Vis detector (Shimadzu, Kyoto, Japan). The Agilent Eclipse XDB-C18 250 × 9.4 mm column (Agilent, Santa Clara, CA) was used for semi-preparative HPLC purification while the Agilent Eclipse XDB-C18 250 × 4.6 mm column (Agilent, Santa Clara, CA) was used for analytical HPLC to check peptide purity. The peptide was lyophilized to obtain a white powder. ESI-HRMS calc for $C_{122}H_{210}N_{36}O_{37}$ [M]$^+$: 2771.5657 found 2771.5587.

## Dot blot

The synthetic peptide was reduced with $NaBH_4$ in sodium carbonate ($Na_2CO_3$) buffer (pH 9.0) at room temperature. The reaction mixture was then spotted on NC membranes. After incubation with 5% non-fat milk for 1 hr, the membrane was incubated with the anti-β-hydroxybutyryllysine antibody (Kbhb) overnight at 4°C. After three washes with TBST (Tris-buffered saline, pH 7.4, 0.1% (vol/vol) Tween 20), the membrane was incubated with a 1:3000 dilution of goat anti-rabbit IgG-HRP antibody at room temperature for 1 hr. Following an additional wash with TBST, chemiluminescent detection was performed using the ECL substrate (Thermo Fisher Scientific, Cat# 32209).

## Preparation of histones and cell lysate

HEK293T cells were cultured in DMEM (Corning, Cat# 10-013-CV) containing 10% FBS (Thermo Fisher Scientific, Cat# A5256801), 1% streptomycin–penicillin (Thermo Fisher Scientific, Cat# 15140122). Cells were treated with lithium acetoacetate (Fisher Scientific, Cat# A1478), sodium β-hydroxybutyrate, valine, or leucine followed by whole cellular proteome extraction or histones extraction. Cell lysates were extracted using M-PER Mammalian Protein Extraction Reagent (Thermo Fisher Scientific, Cat# 78501) and 1% Protease Inhibitor Cocktail III (Thermo Fisher Scientific, Cat# 78425) with gentle sonication. The remaining debris was removed by centrifugation at 13,200 × rpm at 4°C. Histone proteins were extracted with the EpiQuik Total Histone Extraction Kit (Epigentek, Cat# OP-0006-100) according to the manufacturer's instructions. The protein concentration was determined by Bradford assay.

## NaBH$_4$-Kbhb antibody method based western blot analysis

Protein extracts (20 μg of cell lysate or 7 μg of histones) were first reduced by $NaBH_4$ in $Na_2CO_3$ buffer (pH 9.0) at room temperature. Samples after reduction were further fractionated by SDS–PAGE and transferred to an NC membrane. After incubation with 5% non-fat milk in TBST for 1 hr at room temperature, the membrane was incubated with the anti-Kbhb antibody at 1:2000 dilution at 4 °C overnight. Then the membrane was washed three times with TBST and incubated with a 1:3000 dilution of goat anti-rabbit IgG-HRP antibody at room temperature for 1 hr. Following three washes with TBST, the membrane was analyzed by chemiluminescence using the ECL substrate.

## Expression and purification of HATs

HAT1 (20–341), p300 (1287–1666), and MOF (125–458) were expressed and purified as described in the previous work of our lab (*Zhu et al., 2021*; *Yang et al., 2013*). The expression and purification of MOZ protein (pET28a-LIC-MOZ plasmid, Addgene #25181) were done according to the protocol from

Structural Genomics Consortium (SGC) (http://www.thesgc.org/structures/2ozu). Maltose-binding protein (MBP)-MORF (361–716) was expressed and purified as described (*Han et al., 2017*). Plasmids (pET28a-PCAF (493–658) and perceiver HBO1) were transformed into *Escherichia coli* BL21 (DE3)/RIL competent cells with heat-shock method followed by spreading cells on LB-Agar plate with kanamycin or ampicillin. Colonies were picked up and grown in 8 ml of Luria broth medium for 16 hr and then 1 l cultures of Luria broth medium containing kanamycin or ampicillin at 37°C until the $OD_{595nm}$ reached 0.5–0.7. 0.3 mM of isopropyl β-d-1-thiogalactopyranoside was added to induce protein expression at 16°C overnight. Cells were collected by centrifugation and suspended in lysis buffer containing 50 mM sodium phosphate at pH 7.4, 0.25 M NaCl, 5% (vol/vol) glycerol, 0.1% (vol/vol) Triton X, 5 mM imidazole, 2 mM β-mercaptoethanol, and 1 mM phenylmethylsulfonyl fluoride (PMSF). Cells were disrupted with a microfluidic cell disruptor followed by collection of supernatant. After equilibrating nickel-NTA agarose resin with column buffer (20 mM Na-HEPES at pH 8.0, 300 mM NaCl, 10% (vol/vol) glycerol, 30 mM imidazole, and 1 mM PMSF), the protein supernatant was loaded onto the resin. Next, the resin was washed with column buffer twice and washing buffer (20 mM Na-HEPES at pH 8.0, 300 mM NaCl, 10% (vol/vol) glycerol, 70 mM imidazole, and 1 mM PMSF) three times. The proteins on the resin were eluted with elution buffer (20 mM Tris-HCl at pH 8.0, 300 mM NaCl, 10% glycerol, 500 mM imidazole, and 1 mM PMSF), and then dialyzed in the dialysis buffer (25 mM Na-HEPES/Tris-HCl at pH 8.0, 250 mM NaCl, 10% glycerol, 1 mM DTT) at 4°C overnight. The resultant proteins were concentrated by Millipore centrifugal filter. The protein concentration and purity were determined by using Bradford assay and SDS–PAGE, respectively. The proteins were then aliquoted and stored at −80°C.

## In vitro screening of HATs activities

1 μg of recombinant histone H3 or H4 was incubated with individual HAT enzymes and 50 μM acetyl-CoA or acetoacetyl-CoA in a HAT reaction buffer (50 mM HEPES-Na and 0.1 mM EDTA-Na, pH 8.0) at 30°C for 1 hr. The reaction mixture was reduced by $NaBH_4$, followed by the addition of SDS sample buffer. The levels of acetylation and acetoacetylation were assessed by western blot. To confirm the activities of HATs in vitro, synthetic histone peptides H3 (1–20) or H4 (1–20) (the sequence of H3 (1–20): Ac-ARTKQTARKSTGGKAPRKQL; the sequence of H4 (1–20): Ac-SGRGKGGKGLGKG-GAKRHRK) were used as acyl acceptor substrates to allow enzymatic transfer of acetoacetyl- group by individual HAT in the same reaction conditions. The reaction products were detected and validated by MALDI-MS.

## In vivo validation of HAT, HDAC3, AACS, SCOT, and HMGCR activities

HEK293T cells or HepG2 cells were cultured and transfected with pCMVβ-p300-myc (Addgene, Cat# 30489), flag-HDAC3 (Addgene, Cat# 13819), AACS (OriGene Technologies, Cat# RC206247), SCOT (OriGene Technologies, Cat# RC203764), or HMGCR (Addgene, Cat# 86085) by using Lipofectamine 3000 Transfection Reagent (Thermo Fisher, Cat# L3000008). The cells were subsequently treated with 20 mM lithium acetoacetate for 24 hr, after which histones were extracted. Additional experiments were conducted by treating cells with acetohydroxamic acid or lovastatin in combination with lithium acetoacetate to study SCOT or HMGCR activity. Kacac levels were analyzed by western blot using the procedures aforementioned.

## Molecular docking

The referred proteins were imported into Maestro version 13.6.121 and subsequently prepared with filled in missing side chains in the preprocessing stage and optimized hydrogen bond assignments using PROPKA at pH 7.4. Imported acyl-CoAs were prepared with predetermined stereocenters with the generated states at pH 7.0 ± 2.0. The receptor grid was then generated by selecting key residues around the catalytic site and then specified by either maintaining key hydrogen bonding residues or through the use of a core constraint with the co-crystallized ligand. Lastly, ligand docking of the acyl-CoAs applied the constraints used in the receptor grid using extra precision sampling to yield the modeled poses of the acyl CoAs.

## Trypsin digestion of cell lysate

Extracted proteins were reduced with 50 mM $LiBD_4$ in $Na_2CO_3$ buffer (pH 9.0) for 6 hr at room temperature. Whole proteome was precipitated by cold acetone overnight. Proteins were redissolved

in 50 mM NH$_4$HCO$_3$ buffer (pH 8.0), followed by reduction with 20 mM DTT for 1.5 hr at 37°C, and alkylation with 40 mM iodoacetamide for 30 min at room temperature in darkness. Excess iodoacetamide was blocked by 10 mM DTT. Trypsin (Thermo Fisher Scientific, Cat# 90058) was added at a 1:50 trypsin-to-protein mass ratio for protein digestion overnight at 37°C. The enzymatic reaction was quenched by boiling the sample. Tryptic peptides were dried in a SpeedVac system (Thermo Fisher Scientific).

### Peptide immunoprecipitation

Peptides were redissolved in NETN buffer (1 mM EDTA, 50 mM Tris-HCl, 100 mM NaCl, 0.5% NP-40, pH 8.0). Pan anti-Kbhb antibody was first conjugated to Protein A/G plus agarose (Santa Cruz Biotechnology Inc, Cat# sc-2003) and then incubated with tryptically digested peptides with gentle agitation overnight at 4°C. The beads were then washed three times with NETN buffer, twice with ETN buffer (50 mM Tris-HCl pH 8.0, 100 mM NaCl, 1 mM EDTA), and twice with water. Peptides were eluted from the beads with 0.15% TFA and dried by SpeedVac system.

### HPLC–MS/MS analysis

The resulting peptides were dissolved in 0.1% formic acid in water and loaded onto a commercial C18 column Acclaim PepMap RSLCnano, 75 μm × 15 cm, 3 μm particle size (Thermo Fisher Scientific, Waltham, MA). The loaded peptides were separated using a gradient of 5–80% HPLC buffer B (0.1% formic acid in 80% acetonitrile, vol/vol) in buffer A (0.1% formic acid in water, vol/vol) at a flow rate of 300 nl/min over 180 min by Dionex Ultimate 3000 RSLCnano system (Thermo Fisher Scientific, Waltham, MA). The samples were analyzed by an Eclipse tribrid orbitrap mass spectrometer (Thermo Fisher Scientific, Waltham, MA). In positive ion mode, a 120,000 resolution full mass scan was collected, followed by data-dependent MS/MS using 28% higher collision energy fragmentation at 30,000 resolution with a cycle time of 3 s. Charge state screening was enabled and precursors with a charge of +1, or an unknown charge were excluded. A dynamic exclusion duration of 60 s was enabled (*Shajahan et al., 2023*). A lock mass correction was also applied using a background ion (*m/z* 445.12002).

### Protein sequence database searching

The acquired MS/MS data was searched by Byonic (Protein Metrics; v4.1). All the data were searched against reviewed UniProt Human protein database (20,433 entries, http://www.uniprot.org) with decoy. Trypsin was specified as cleavage enzyme allowing a maximum of two missing cleavages. Cysteine carbamidomethylation was specified as fixed modification. Methionine oxidation, lysine acetylation, lysine methylation, lysine β-hydroxybutyrylation, deuterated lysine β-hydroxybutyrylation (DKbhb), lysine acetoacetylation, lysine propionylation, and lysine butyrylation were included as variable modifications. FDR thresholds for protein, peptide, and modification site were specified at 1%. Peptides with a Byonic PEP 2D value lower than 0.001, as well as those with PEP 2D values between 0.1 and 0.001 after manual confirmation, were retained for subsequent analysis.

### Bioinformatics analysis

Sequence preference motif was generated by iceLogo, utilizing the human proteome as the background (*Colaert et al., 2009*). Mutations and recorded binding sites were extracted from the UniProt database (http://www.uniprot.org). GO analysis was performed using PANTHER (version 18.0) (*Mi et al., 2016*). Enrichment analysis for KEGG pathway was carried out using the GOstats package along with a hypergeometric test in R (*Falcon and Gentleman, 2007*). Protein complex analysis was performed by using the manually curated CORUM protein complex database for all mammals using a hypergeometric test (*Ruepp et al., 2010*). Protein complexes enriched in the Kacac proteome were visualized in Cytoscape (v3.10.1) (*Shannon et al., 2003*).

### RNA-seq analysis

Total RNAs were extracted from control and lithium acetoacetate-treated (20 mM for 24 hr) HEK293T cells using the RNeasy Plus Mini Kit (QIAGEN, Cat# 74134). Three biological replicates were performed for each condition. Isolated RNA sample quality was assessed by High Sensitivity RNA Tapestation (Agilent Technologies Inc, California, USA) and quantified by AccuBlue Broad Range RNA Quantitation

assay (Biotium, California, USA). Paramagnetic beads coupled with oligo d(T)25 were combined with total RNA to isolate poly(A)+ transcripts based on NEBNext Poly(A) mRNA Magnetic Isolation Module manual (New England BioLabs Inc, Massachusetts, USA). Prior to first strand synthesis, samples were randomly primed (5′ d(N6) 3′ [N = A,C,G,T]) and fragmented based on the manufacturer's recommendations. The first strand was synthesized with the Protoscript II Reverse Transcriptase with a longer extension period, approximately 30 min at 42°C. All remaining steps for library construction were used according to the NEBNext Ultra II Directional RNA Library Prep Kit for Illumina (New England BioLabs Inc, Massachusetts, USA). Final libraries quantity was assessed by Qubit 2.0 (Thermo Fisher, Massachusetts, USA) and quality was assessed by TapeStation D1000 ScreenTape (Agilent Technologies Inc, California, USA). Final library size was about 450 bp with an insert size of about 300 bp. Illumina 8-nt dual-indices were used. Equimolar pooling of libraries was performed based on QC values and sequenced on an Illumina NovaSeq X Plus 10B (Illumina, California, USA) with a read length configuration of 150 PE for 40 M PE reads per sample (20 M in each direction). FastQC (version v0.12.1) was employed to check the quality of raw reads. Trimmomatic (version v0.39) was applied to cut adaptors and trim low-quality bases with default settings. STAR Aligner (version 2.7.10b) was used to align the reads. The package of Picard tools (version 3.0.0) was applied to mark duplicates of mapping. StringTie (version 2.2.1) was used to assemble the RNA-seq alignments into potential transcripts. FeatureCounts (version 2.0.6) or HTSeq (version 2.0.3) was used to count mapped reads for genomic features such as genes, exons, promoters, gene bodies, genomic bins, and chromosomal locations. DESeq2 (version 1.40.2) was employed to process the differential gene expression analysis. The list of significance was established by setting the fold change threshold at a level of 1.5 and adjusted $p < 0.05$. The GO, GSEA, and KEGG enrichment analysis were analyzed via the ClusterProfiler package and Molecular Signatures Database (MSigDB).

## Acknowledgements

We thank the Proteomics and Mass Spectrometry facility (PAMS) at UGA for the MS support. Proteomic analysis was performed at the Complex Carbohydrate Research Center and was supported in part by the National Institutes of Health (NIH)-funded R24 grant [R24GM137782 to Parastoo Azadi]. The Eclipse mass spectrometer used in the modified peptide analysis was supported by GlycoMIP, a National Science Foundation Materials Innovation Platform funded through Cooperative Agreement DMR-1933525. We are thankful to the National Institutes of Health [NIH 1R35GM149230] and the National Science Foundation [NSF 2203942] for grant support (PI Zheng).

## Additional information

### Funding

| Funder | Grant reference number | Author |
|---|---|---|
| National Institutes of Health | NIH 1R35GM149230 | Y George Zheng |
| National Institutes of Health | R24GM137782 | Parastoo Azadi |
| National Science Foundation | DMR-1933525 | Parastoo Azadi |
| National Science Foundation | NSF 2203942 | Y George Zheng |

The funders had no role in study design, data collection, and interpretation, or the decision to submit the work for publication.

### Author contributions

Qianyun Fu, Conceptualization, Data curation, Software, Formal analysis, Validation, Investigation, Visualization, Methodology, Writing – original draft, Project administration, Writing – review and editing; Terry Nguyen, Data curation, Software, Formal analysis, Investigation, Visualization, Writing

– original draft, Writing – review and editing; Bhoj Kumar, Data curation, Investigation, Visualization, Writing – original draft, Writing – review and editing; Parastoo Azadi, Resources, Formal analysis, Validation; Y George Zheng, Conceptualization, Resources, Formal analysis, Supervision, Funding acquisition, Validation, Methodology, Writing – original draft, Project administration, Writing – review and editing

### Author ORCIDs
Qianyun Fu ⓘ https://orcid.org/0000-0002-5099-2736
Terry Nguyen ⓘ https://orcid.org/0009-0003-2347-1390
Bhoj Kumar ⓘ https://orcid.org/0000-0001-8311-7025
Y George Zheng ⓘ https://orcid.org/0000-0001-7116-3067

Reviewer #3 (Public review): https://doi.org/10.7554/eLife.104123.4.sa1
Author response https://doi.org/10.7554/eLife.104123.4.sa2

---

## Additional files

### Supplementary files
MDAR checklist

### Data availability
All data generated or analyzed during this study are included in the manuscript file. Source data files have been provided.

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
