## [Editor Report · eLife Assessment]

This **useful** study reports a method to detect and analyze a novel post-translational modification, lysine acetoacetylation (Kacac), finding it regulates protein metabolism pathways. The study unveils epigenetic modifiers involved in placing this mark, including key histone acetyltransferases such as p300, and concomitant HDACs, which remove the mark. Proteomic and bioinformatics analysis identified many human proteins with Kacac sites, potentially suggesting broad effects on cellular processes and disease mechanisms. The data presented are **solid** and the study will be of interest to those studying protein and metabolic regulation.

---

## [Referee Report · Reviewer #3 (Public review)]

Summary:

This paper presents a timely and significant contribution to the study of lysine acetoacetylation (Kacac). The authors successfully demonstrate a novel and practical chemo-immunological method using the reducing reagent NaBH4 to transform Kacac into lysine β-hydroxybutyrylation (Kbhb).

Strengths:

This innovative approach enables simultaneous investigation of Kacac and Kbhb, showcasing their potential in advancing our understanding of post-translational modifications and their roles in cellular metabolism and disease.

Weaknesses:

The study lacks supporting in vivo data, such as gene knockdown experiments, to validate the proposed conclusions at the cellular level.

---

## [Author Response]

The following is the authors’ response to the previous reviews

**Public Reviews:**

**Reviewer #2 (Public review):**
In the manuscript by Fu et al., the authors developed a chemo-immunological method for the reliable detection of Kacac, a novel post-translational modification, and demonstrated that acetoacetate and AACS serve as key regulators of cellular Kacac levels. Furthermore, the authors identified the enzymatic addition of the Kacac mark by acyltransferases GCN5, p300, and PCAF, as well as its removal by deacetylase HDAC3. These findings indicate that AACS utilizes acetoacetate to generate acetoacetyl-CoA in the cytosol, which is subsequently transferred into the nucleus for histone Kacac modification. A comprehensive proteomic analysis has identified 139 Kacac sites on 85 human proteins. Bioinformatics analysis of Kacac substrates and RNA-seq data reveal the broad impacts of Kacac on diverse cellular processes and various pathophysiological conditions. This study provides valuable additional insights into the investigation of Kacac and would serve as a helpful resource for future physiological or pathological research.The authors have made efforts to revise this manuscript and address my concerns. The revisions are appropriate and have improved the quality of the manuscript.

We appreciate the constructive and thoughtful feedbacks, which have been invaluable in enhancing the quality of our manuscript.

**Reviewer #3 (Public review):**
Summary:This paper presents a timely and significant contribution to the study of lysine acetoacetylation (Kacac). The authors successfully demonstrate a novel and practical chemoimmunological method using the reducing reagent NaBH4 to transform Kacac into lysine βhydroxybutyrylation (Kbhb).

Thank you for the positive and insightful comments.

Strengths:This innovative approach enables simultaneous investigation of Kacac and Kbhb, showcasing its potential in advancing our understanding of post-translational modifications and their roles in cellular metabolism and disease.

We are grateful for the reviewer’s comments, which has contributed to enhancing the quality of our study.

Weaknesses:The experimental evidence presented in the article is insufficient to fully support the authors' conclusions. In the in vitro assays, the proteins used appear to be highly inconsistent with their expected molecular weights, as shown by Coomassie Brilliant Blue staining (Figure S3A). For example, p300, which has a theoretical molecular weight of approximately 270 kDa, appeared at around 37 kDa; GCN5/PCAF, expected to be ~70 kDa, appeared below 20 kDa. Other proteins used in the in vitro experiments also exhibited similarly large discrepancies from their predicted sizes. These inconsistencies severely compromise the reliability of the in vitro findings. Furthermore, the study lacks supporting in vivo data, such as gene knockdown experiments, to validate the proposed conclusions at the cellular level.

We appreciate the reviewer’s comments. In the biochemical assays, we used the expressed catalytic domains of HATs—rather than the full-length proteins for activity testing. Specifically, the following constructs were expressed and purified: p300 (1287– 1666), GCN5 (499-663), PCAF (493-658), MOF (125-458), MOZ (497-780), MBP-MORF (361-716), Tip60 (221-512), HAT1 (20-341), and HBO1 (full length). This resulted in the observed discrepancies in molecular weight in Figure S3A compared to the expected fulllength weights.

Although a recent study (PMID: 37382194) reported the acetoacetyltransferase activities of p300 and GCN5 in cells, we recognize that additional knockdown experiments would be necessary to substantiate their contributions to in vivo Kacac generation and to explore the functional roles of Kacac in an enzyme-specific context. We plan to address these kinds of research issues in our future work.